# VIDEORFT: Incentivizing Video Reasoning Capability in MLLMs via Reinforced Fine-Tuning

Qi Wang[1,2], Yanrui Yu[2], Ye Yuan[2,3], Rui Mao[4], Tianfei Zhou[2,3,*]

[1]Beijing Institute of Technology, Zhuhai   [2]Beijing Institute of Technology
[3]State Key Laboratory of Environment Characteristics and Effects for Near-space
[4]Shenzhen University
https://github.com/QiWang98/VideoRFT

## Abstract

Reinforcement fine-tuning (RFT) has shown great promise in achieving human-level reasoning capabilities of Large Language Models (LLMs), and has recently been extended to MLLMs. Nevertheless, reasoning about videos, which is a fundamental aspect of human intelligence, remains a persistent challenge due to the complex logic, temporal and causal structures inherent in video data. To fill this gap, we propose **VIDEORFT**, a novel approach that extends the RFT paradigm to cultivate human-like video reasoning capabilities in MLLMs. **VIDEORFT** follows the standard two-stage scheme in RFT: supervised fine-tuning (SFT) with chain-of-thought (CoT) annotations, followed by reinforcement learning (RL) to improve generalization. A central challenge to achieve this in the video domain lies in the scarcity of large-scale, high-quality video CoT datasets. We address this by building a multi-expert-driven, cognition-inspired CoT curation pipeline. First, we devise a cognition-inspired prompting strategy to elicit a reasoning LLM to generate preliminary CoTs based solely on rich, structured, and literal representations of video content. Subsequently, these CoTs are revised by a MLLM conditioned on the actual video, ensuring visual consistency and reducing visual hallucinations. This pipeline results in two new datasets, *i.e.*VideoRFT-CoT-102K for SFT and VideoRFT-RL-310K for RL. To further strengthen the RL phase, we introduce a novel semantic-consistency reward that explicitly promotes the alignment between textual reasoning and visual evidence. This reward encourages the model to produce coherent, context-aware reasoning outputs grounded in visual input. Extensive experiments show that **VIDEORFT** achieves state-of-the-art performance on six video reasoning benchmarks.

## 1 Introduction

The ability to reason about complex videos lies at the core of human cognitive development [36]. Humans, even infants, exhibit a remarkable capacity to understand videos – recognizing what has happened, inferring what will happen next, and explaining why events occur. Replicating this capability in AI systems has become a central goal in video understanding, and has been extensively studied in the field of computer vision over the past decade [4, 29, 47, 57]. Despite the progress, most AI models remain limited to perceptual-level understanding and struggle to reason about video content with the depth, efficiency, and interpretability that are characteristic of human cognition.

Recently, the rapid development of video MLLMs, such as Video-ChatGPT [28], VILA [24], Video-LLaVA [23], has significantly advanced the capabilities of video understanding. However, these

---

[*]Corresponding Author

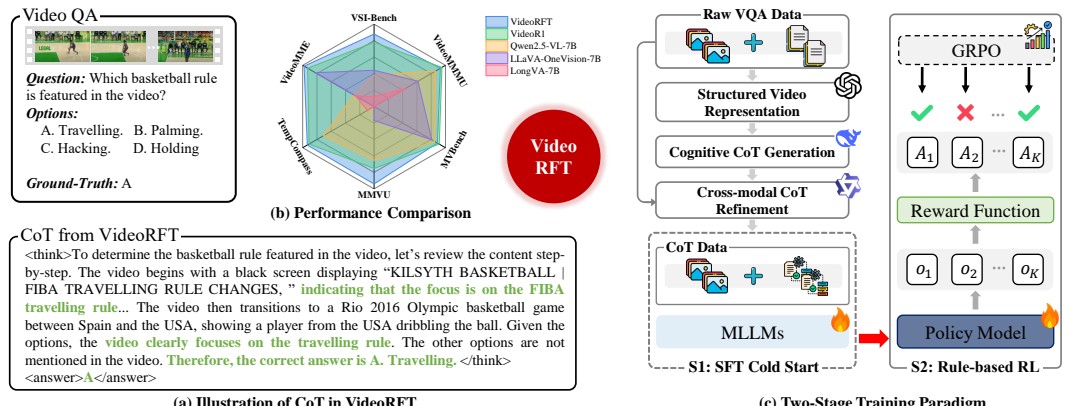

Figure 1: **Overview of VIDEORFT**. (a) An example of CoT derived from **VIDEORFT**. (b) **VIDEORFT** achieves leading performance in six datasets. (c) The two-stage RFT underpins the training of **VIDEORFT**.

models are predominantly answer-driven, *i.e.*, they produce answers without explicitly revealing the reasoning process. VoT [8] overcomes this by introducing a human-like reasoning framework that structures the task of video reasoning into five predefined steps using fixed templates. Nevertheless, such a rigid, template-based approach stands in contrast to the flexibility of human cognition, which enables adaptive reasoning processes based on perceptual inputs [2, 6].

In contrast, the very recent advancements, *e.g.*, OpenAI-o1 [17], DeepSeek-R1 [11], and Kimi-1.5 [38], have shifted focus towards building LLMs that think before answering. These models show strong proficiency in interpreting complex problems, performing multi-step reasoning, and ultimately arriving at correct answers. A key enabler to such capabilities is reinforcement fine-tuning (RFT) [27], which typically commences with a warm-up phase of supervised fine-tuning using CoTs, and subsequently refines the model through reinforcement learning algorithms (*e.g.*, PPO [32], GRPO [34]). Beyond the language domain, pioneering efforts have extended RFT to MLLMs to enhance image-based capabilities [15, 26, 37, 44, 49, 54], and some works [9, 20, 51] that concurrently with ours, show the potential of RFT in the video domain. However, there is a critical challenge remaining unsolved: current video CoT datasets lack the complexity and granularity necessary for advanced video reasoning, which fundamentally limits the ability of models to emulate human-level cognitive capabilities. Moreover, how to ensure that reasoning outputs are faithfully grounded in visual evidence remains underexplored in these works.

Motivated by the above analysis, we propose **VIDEORFT**, a novel reinforcement fine-tuning framework to incentivize the video reasoning capability in MLLMs (see Fig. 1). To overcome the scarcity of video CoTs, we develop a scalable, cognitively inspired pipeline that integrates multiple expert models to collaboratively construct high-quality video CoT datasets. Specifically, we first employ a specialized MLLM to extract structured textual descriptions from videos, capturing fine-grained visual details. These descriptions are then processed by a reasoning-capable LLM (*e.g.*, DeepSeek-R1), which generates initial CoTs through blind reasoning—relying solely on textual input. However, due to the lack of direct visual grounding, such CoTs often contain inconsistencies and hallucinations [14]. To mitigate this issue, we introduce a cross-modal revision stage, wherein a MLLM refines the initial CoTs by incorporating the original video, ensuring consistency with visual evidence. Based on this pipeline, we construct two large-scale datasets, *i.e.*, VideoRFT-CoT-102K and VideoRFT-RL-310K, which together support the RFT process in **VIDEORFT**.

Furthermore, to strengthen the RL phase, we develop a novel semantic-consistency reward that explicitly enhances the visual faithfulness of reasoning outputs in MLLMs. Our key observation is that the reasoning traces of MLLMs are typically structured into three consecutive parts: *question parsing*, *video describing*, and *abstract reasoning*. While the *question parsing* and *abstract reasoning* components are not necessarily grounded in the visual input, the *video describing* part should be closely aligned with the actual visual semantics. Based on this insight, our semantic-consistency reward measures the alignment between the token representations of the *video description* part and the visual features of the input video. This reward is integrated into the GRPO algorithm to guide MLLMs toward generating visually grounded outputs.

**Contributions of this work.** We propose **VIDEORFT**, a novel framework that extends RFT to MLLMs so as to emulate human-like video reasoning capabilities. To achieve this, we first establish a CoT foundation for video RFT by designing a cognitively inspired pipeline to curate large-scale, high-quality video CoT annotations. Furthermore, we introduce a novel semantic-consistency reward to explicitly guide the reasoning trajectories of MLLMs grounded in visual evidence, which enhances the effectiveness of RFT in cross-modal reasoning. Built on these contributions, **VIDEORFT** favorably outperforms advanced competitors on a series of challenging video reasoning benchmarks.

## 2  VIDEORFT CoT Dataset

We first present the construction of VideoRFT-COT and VideoRFT-RL to support RFT in MLLMs.

### 2.1  Data Collection

We extensively collect video question-answer data covering diverse modalities, task types, and cognitive skills. Given the scarcity of high-quality video data in certain domains (*e.g.*, mathematics, science), we additionally incorporate carefully curated image-based instances. The final dataset contains 310K samples in total, supporting diverse answer formats, including multiple-choice (mc), numerical (num), free-form text generation (free), optical character recognition (ocr), and regression (reg). As shown in Fig. 2, the samples are categorized into five groups according to the type of cognitive skills involved in the reasoning process:

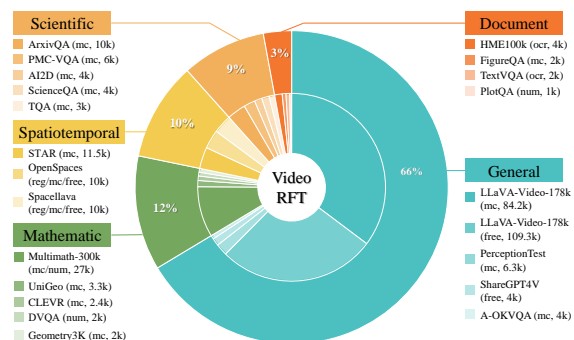

Figure 2: The distribution of data collection.

- **General**: Commonsense reasoning in open-domain temporal and causal contexts.
- **Mathematics**: Symbolic reasoning and spatial alignment for multi-step logic tasks.
- **Science**: Domain-specific reasoning in physics, chemistry, and medicine, emphasizing causal reasoning and conceptual abstraction.
- **Document**: Targets structured visual parsing and information extraction from complex layouts.
- **Spatiotemporal**: Involves motion prediction, spatial transformation, and relational reasoning.

### 2.2  Cognitively Inspired CoT Generation

To enable MLLMs to acquire human-like reasoning abilities, it is essential to construct a high-quality, cognitively grounded video CoT dataset. We propose an automated pipeline for generating such CoT data. As illustrated in Fig. 3, the pipeline comprises three major stages, and *all the prompts used in the pipeline are provided in the supplementary material.*

**Structured Video Representation.** For each video $v$, we generate semantically rich textual descriptions by prompting GPT-4o-mini [16]. The prompt $P_{\text{rep}}$ is carefully crafted to guide the model to (i) summarize video content with a high-level caption, and (ii) produce analytical, frame-level metadata for uniformly sampled video frames. Each frame is structured in a predefined JSON schema that includes timestamped captions and key visual elements such as objects, actions, scenes, spatial relations, and potential interactions. We denote the structured representation of $v$ as $S_v$.

**Cognitively Inspired CoT Generation.** Given the representation $S_v$ and a corresponding question $q$, we invoke a LLM, *i.e.*, DeepSeek-R1 to answer the question and extract its step-by-step reasoning outputs as the initial CoT, *i.e.*, $\text{CoT}_v^{(0)}$:

$$\text{CoT}_v^{(0)} = \text{LLM}(q, S_v, P_{\text{cog}}). \tag{1}$$

Here, $P_{\text{cog}}$ denotes a composite prompt consisting of five sub-prompts *i.e.*, $P_{\text{cog}} = [p_s, p_t, p_a, p_v, p_r]$, each guiding a distinct stage of the reasoning trajectory in a manner that mimics human cognitive processing. Concretely, these sub-prompts are defined as follows. (1) *Simulated observation prompt*

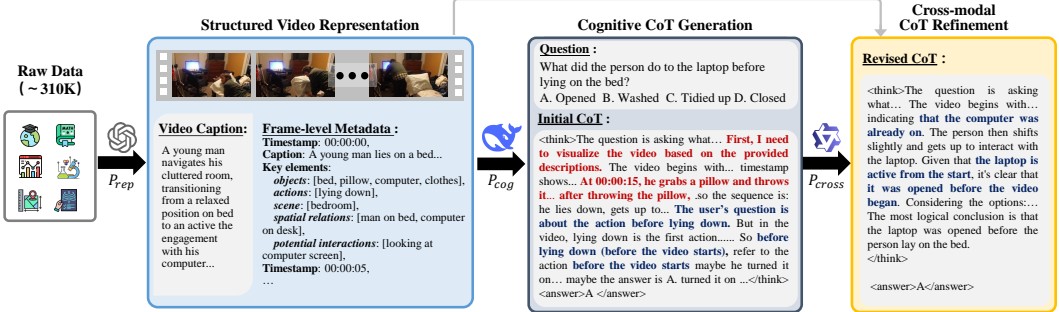

Figure 3: Illustration of the pipeline for cognitively inspired CoT generation.

$(p_s)$: Instruct the model to simulate viewing the entire video and form an initial high-level understanding. (2) *Task understanding prompt* $(p_t)$: Encourage analysis of the question $q$ to infer the task type (*e.g.*, fact, reason, causal relationship). (3) *Selective focus prompt* $(p_a)$: Direct attention to specific temporal segments of the video relevant to $q$. (4) *Visual reasoning prompt* $(p_v)$: Ground the reasoning process in visual content, encouraging analysis over objects, actions, spatial-temporal relations, and event transitions. (5) *Reflective answering prompt* $(p_r)$: Guide the model to derive the final answer, optionally incorporating self-verification or reflection to ensure reasoning quality.

**Cross-modal CoT Refinement.** A key limitation in the initial CoTs is that they might suffer from visual hallucinations due to the lack of visual cues in Eq. 1. To resolve this issue, we introduce a cross-modal refinement strategy to revise the CoT so that it aligns better with the actual video input. Specifically, we prompt a MLLM, *i.e.*, Qwen2.5-VL [1], to compare the initial CoT with the video $v$, identify inconsistencies, and perform necessary revisions:

$$\text{CoT}_v = \text{MLLM}(v, \text{CoT}_v^{(0)}, P_{\text{cross}}). \tag{2}$$

Here the prompt $P_{\text{cross}}$ is designed to guide the MLLM to: (i) verify the cross-modal alignment of $\text{CoT}_v^{(0)}$ with the content of video $v$, (ii) localize and explain any visual-textual inconsistencies, and (iii) revise the CoT to enhance visual grounding while preserving its original logical structure.

Finally, we apply a filtering stage to ensure the factual correctness of the resulting CoT annotations. For structured tasks with clear ground-truth labels, we directly exclude samples with incorrect final answers. For open-ended tasks, we remove samples exhibiting low semantic consistency (measured by CLIP [31]) between the generated answer and the reference answer. This filtering process ensures that the resulting CoT dataset maintains high quality and factual reliability. After filtering, 102K high-confidence samples are retained from the initial pool of 310K, forming VideoRFT-CoT-102K for supervised fine-tuning.

### 2.3 Data Analysis

Fig. 4 presents a comparative analysis of the CoTs in our proposed VideoRFT-CoT-102K and Video-R1 [9]. As shown in Fig. 4 (a), the CoTs in our dataset exhibit a broader distribution and longer average token length compared to those in Video-R1 (Fig. 4(c)), indicating that our VideoRFT-CoT-102K contains more elaborate, fine-grained, and nuanced reasoning processes. Additionally, the word cloud in Fig. 4 (b) reveals that CoTs in VideoRFT-CoT-102K are dominated by dynamic, video-centric concepts such as "*video*", "*main*", "*happen*", and "*first*". The lexical profile reflects an emphasis on narrative structure and temporal progression, which are the key characteristics of complex video understanding. In contrast, Video-R1 (Fig. 4 (d)) features frequent refer-

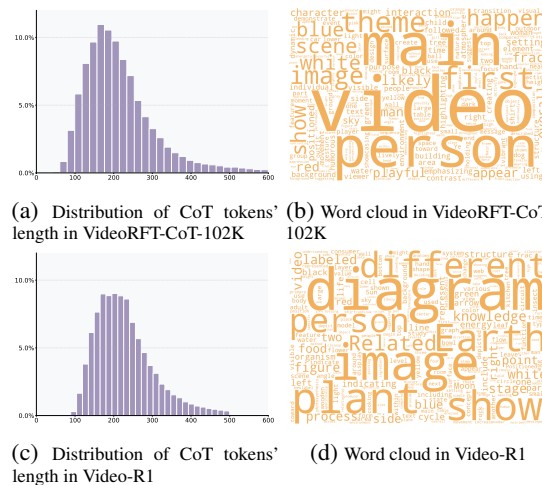

(a) Distribution of CoT tokens' length in VideoRFT-CoT-102K

(b) Word cloud in VideoRFT-CoT-102K

(c) Distribution of CoT tokens' length in Video-R1

(d) Word cloud in Video-R1

Figure 4: Comparison of CoT dataset in VideoRFT-CoT-102K and Video-R1.

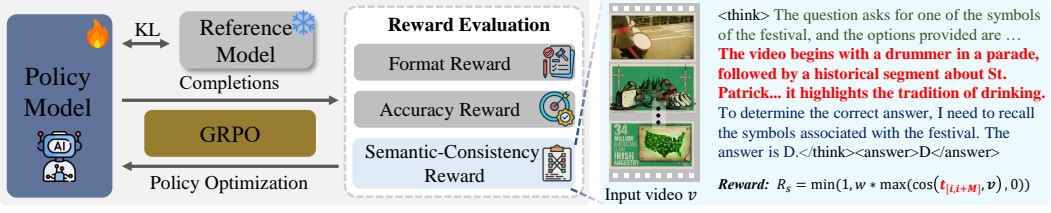

(a) Rule-based Reinforcement Learning      (b) Semantic-Consistency Reward

Figure 5: Illustrations of (a) rule-based RL, and (b) the computation of semantic-consistency reward $R_s$. The reasoning outputs are color-coded to highlight *question parsing* (green), *video description* (red) and *abstract reasoning* (blue). Only the red part is involved in the computation of $R_s$ (see §3.2).

ences to static or declarative content, *e.g.*, "*diagram*", "*image*", "*plant*", and "*Earth*", suggesting a stronger bias toward factual descriptions rather than deep reasoning. These results highlight that VideoRFT-CoT-102K offers greater expressiveness in reasoning depth and aligns more closely with the demands of real-world video reasoning tasks. Hence, it provides a better foundation for training video MLLMs with advanced reasoning capabilities.

# 3   Video Reinforcement Fine-Tuning

This section presents our approach for video reinforcement fine-tuning as shown in Fig. 5. We first provide a brief overview of Group Relative Policy Optimization (GRPO) [34] in §3.1, and then elaborate on the proposed rule-based reward for efficient reinforcement fine-tuning in §3.2.

## 3.1   Group Relative Policy Optimization

GRPO [34] is a computationally efficient rule-based RL algorithm designed specifically for training large reasoning models. Unlike traditional RL methods such as PPO [32], which require four models (policy, value, reward, and reference), GRPO simplifies the approach by eliminating the value model, significantly reducing memory requirements and training complexity. GRPO operates by generating $K$ candidate responses $\{o_1, o_2, \ldots, o_K\}$ for each query $q$. These responses are then evaluated using defined reward functions, yielding rewards $\{r_1, r_2, \ldots, r_K\}$. Afterwards, these rewards are normalized to calculate the advantage $A_i$ for each response as:

$$A_i = \frac{r_i - \mathtt{mean}(\{r_1, r_2, \ldots, r_K\})}{\mathtt{std}(\{r_1, r_2, \ldots, r_K\})}, \tag{3}$$

where $\mathtt{mean}$ and $\mathtt{std}$ denote the mean and standard deviation of the rewards, respectively. Subsequently, the model is optimized through maximization of the following objective:

$$\mathcal{J}_{\text{GRPO}}(\theta) = \mathbb{E}_{[q, \{o_i\}]} \frac{1}{K} \sum_{i=1}^{K} \left[ \min \left( \frac{\pi_\theta}{\pi_{\theta_{\text{old}}}} A_i, \text{clip} \left( \frac{\pi_\theta}{\pi_{\theta_{\text{old}}}}, 1 - \epsilon, 1 + \epsilon \right) A_i \right) - \beta \mathbb{D}_{\text{KL}}(\pi_\theta || \pi_{\text{ref}}) \right], \tag{4}$$

where $\theta$ denotes model parameters to be updated, $\pi_\theta$ and $\pi_{\theta_{\text{old}}}$ are the current and old policy model, $\pi_{\text{ref}}$ indicates the reference policy, $\beta$ is the KL divergence regularization coefficient, and $\epsilon$ is a regularization coefficient that prevents the policy from deviating too far from the reference model.

## 3.2   Rule-based Reward Modeling in VIDEORFT

The rewards in Eq. 3 are derived from rule-based reward functions, which represent a foundational step in rule-based RL by simply evaluating whether model predictions exactly match ground-truth answers. Two highly resilient rule-based rewards are the Format Reward and Accuracy Reward, which are consistently utilized in DeepSeek-R1 and its follow-ups. However, in the context of cross-modal reasoning, these rewards are insufficient to provide explicit guidance to MLLMs towards visually grounded reasoning. To address this limitation, we introduce a semantic-consistency reward, which enforces the grounding of generated reasoning content in the visual input.

### 3.2.1 Semantic-Consistency Reward

The reward is motivated by the observation that the reasoning trace generated by video MLLMs typically consists of three distinct parts, *i.e.*, *question parsing*, *video describing* and *abstract reasoning*, as shown in Fig. 5. Among them, the *video describing* stage represents the model's understanding of visual content, which is the foundation for subsequent reasoning. Therefore, the reward is designed to selectively promote alignment between this stage and the input video.

Formally, to isolate the *video describing* sentence from the generated response, we apply a regular expression to locate the first full stop. Empirically, the text following this full stop corresponds to the model's interpretation of visual content. From this point, we extract a fixed-length span of $M$ tokens, denoted $t_{[i,i+M]}$, and encode it using SigLIP [48]'s text encoder: $\boldsymbol{t}_{[i,i+M]} = \text{SigLIP}_{\text{text}}(t_{[i,i+M]})$. Additionally, we uniformly sample $F$ frames $\{v^{(0)}, \ldots, v^{(F-1)}\}$ from video $v$, and compute the visual representation of each frame $v^{(i)}$ via SigLIP's image encoder: $\boldsymbol{v}^{(i)} = \text{SigLIP}_{\text{image}}(v^{(i)})$. Then the final video representation $\boldsymbol{v}$ is naturally obtained by averaging the frame embeddings: $\boldsymbol{v} = \frac{1}{F} \sum_{i=0}^{F} \boldsymbol{v}^{(i)}$. We then define the semantic-consistency reward as:

$$R_s = \min\left(1, w \times \max\left(\cos\left(\boldsymbol{t}_{[i,i+M]}, \boldsymbol{v}\right), 0\right)\right), \tag{5}$$

where $\cos(\cdot, \cdot)$ denotes cosine similarity, and $w = 2$ is a scaling constant. The $\max(\cdot, 0)$ ensures non-negativity of the reward, while the $\min(\cdot, 1)$ stabilizes training by bounding the reward. This stage-aware formulation allows us to reward only the part of reasoning tied to visual comprehension, without penalizing abstract reasoning that appropriately extends beyond the visual scope. The result is enhanced semantic fidelity, reduced hallucinations, and improved alignment during RL.

### 3.3 Overall Reward

**VIDEORFT** uses three types of rewards for RL:

- **Format Reward.** During RL, we incorporate the widely-used format reward to guide the model in generating its reasoning process and final answer in a structured format. This reward, denoted as $R_f$, ensures that the model's output adheres to a predefined structure: the reasoning process must be enclosed within `<think>`...`</think>` tags, and answers within `<answer>`...`</answer>` tags. Compliance is verified via regular expression matching, and a binary reward is assigned accordingly.
- **Accuracy Reward.** To provide reliable supervision across heterogeneous tasks, we adopt task-specific accuracy metrics: Exact Match for multiple-choice and numerical questions, ROUGE for open-ended generation, Word Error Rate for OCR tasks, and a scaled relative accuracy for regression problems. These tailored evaluations ensure the reward $R_a$ aligns with each task.
- **Semantic-Consistency Reward.** The semantic-consistency reward $R_s$, defined in Eq. 5, promotes alignment between the reasoning text and the input visual information.

The overall reward $R$ for a sample is computed as follows:

$$R = R_f + R_a + \mathbb{1}[R_a > 0] \cdot R_s, \tag{6}$$

where $\mathbb{1}[R_a > 0]$ is the indicator function that returns 1 if $R_a > 0$ and 0 otherwise. This indicator function acts as a gate to ensure $R_s$ is activated only when $R_a$ is non-zero, thus avoiding the reinforcement of semantically plausible but factually incorrect reasoning.

## 4 Experiment

### 4.1 Experimental Setup

**Benchmark and Metric.** Following previous works [9, 18, 51], we evaluate our approach on six video reasoning and understanding benchmarks: VSI-Bench [43], VideoMMMU [13], MMVU [55], MVBench [19], TempCompass [25], and VideoMME [10], covering spatial reasoning, knowledge-intensive video QA, temporal logic, and general video understanding. Following conventions, we only use the subset of multiple-choice samples in MMVU, and VideoMME is evaluated without subtitles. Average accuracy is adopted as the evaluation metric.

Table 1: **Performance Comparison.** The best results are highlighted in **bold**. [†]: Results are obtained using larger input resolutions, up to $768\times28\times28$ and 768 sampled frames, while ours are $256\times28\times28$ and 32.

| Model | Pub | Video Reasoning | | | Video Understanding | | |
|---|---|---|---|---|---|---|---|
| | | VSI. | VideoMMMU | MMVU | MV. | TempC. | VideoMME |
| ● **Proprietary Models** | | | | | | | |
| GPT-4o [16] | – | 34.0 | 61.2 | 75.4 | - | - | 71.9 |
| ● **Open-Source Models** | | | | | | | |
| LLaMA-VID [21] | ECCV 24 | - | - | - | 41.9 | 45.6 | - |
| ShareGPT4Video [3] | NeurIPS 24 | - | - | - | 51.2 | - | 39.9 |
| VideoLLaMA2 [5] | arXiv 24.06 | - | - | 44.8 | 54.6 | - | 47.9 |
| LongVA-7B [50] | TMLR 24 | 29.2 | 23.9 | - | - | 56.9 | 52.6 |
| VILA-1.5-8B [24] | CVPR 24 | 28.9 | 20.8 | - | - | 58.8 | - |
| LLaVA-OneVision-7B [18] | TMLR 24 | 32.4 | 33.8 | 49.2 | 56.7 | - | 58.2 |
| mPLUG-Owl3-8B [46] | ICLR 25 | - | - | - | 54.5 | - | 53.5 |
| Qwen2.5-VL-7B [1] | arXiv 25.02 | 31.8 | 47.4 | 61.3 | 59.4 | 69.2 | 52.8 |
| ● **Concurrent R1-based Models** | | | | | | | |
| Video-R1 [9] | arXiv 25.03 | 35.8 | **52.3** | 63.8 | 63.9 | 73.2 | 59.3 |
| TinyLLaVA-Video-R1 [51] | arXiv 25.04 | - | - | 46.9 | - | 49.5 | 46.6 |
| VideoChat-R1 [20] | arXiv 25.04 | - | - | - | 67.9[†] | - | - |
| **VIDEORFT** | – | **36.8** | 51.1 | **68.5** | 62.1 | **73.7** | **59.8** |

**Model Training.** We follow the RFT to train **VIDEORFT** in two stages: the warm-up SFT stage and the rule-based RL stage. Specifically, the SFT stage equips the model with the ability to generate correct responses for diverse questions, and is trained based on VideoRFT-CoT-102K. The rule-based RL stage is based on VideoRFT-RL-310K using the reward in Eq. 6 to optimize structured reasoning and ensure factual validity. The RL training is implemented using the HuggingFace TRL library [39], and our codebase is built upon Open-R1 [7].

**Implementation Details.** We use Qwen2.5-VL-7B [1] as the base model and train **VIDEORFT** on 8 NVIDIA A800 GPUs, with 80GB each. For efficiency, the video input is limited to 16 frames, with each frame processed into $128 \times 28 \times 28$ resolution during training, where $28 \times 28$ is the size of each image patch, and 128 denotes the number of patches. During inference, we increase the number of frames to 32 and the resolution to $256 \times 28 \times 28$. For efficiency, we use a lightweight version of SigLIP with 400M parameters in computing the semantic-consistency reward. The entire model is trained for one epoch of SFT followed by 1K steps of RL.

## 4.2 Main Result

As shown in Table 1, we compare **VIDEORFT** against a variety of baselines, including proprietary models (*i.e.*, GPT-4o [16]), Open-Source MLLMs (*e.g.*, Qwen2.5-VL [1], VILA [24], LongVA [50]), and contemporaneous models (*e.g.*, Video-R1 [9], TinyLLaVA-Video-R1 [51], VideoChat-R1 [20]).

Several key observations can be drawn from the results. First, compared to our base model, *i.e.*, Qwen2.5-VL-7B, **VIDEORFT** achieves significant improvements across all six benchmarks, *e.g.*, **+5.0%** on VSI-Bench, **+7.2%** on MMVU, and **+7.0%** on VideoMME. This demonstrates the effectiveness of our approach in incentivizing video reasoning capabilities in MLLMs. Moreover, **VIDEORFT** consistently outperforms all non-RL Open-Source MLLMs. Second, **VIDEORFT** surpasses the proprietary GPT-4o on VSI-Bench by **+2.8%**, highlighting the strong potential of RFT in bridging the performance gap with closed-source models in the task of video reasoning. Third, when compared to contemporaneous works, our model delivers the best overall performance, **ranking first on four out of six** benchmarks. This validates the superiority and generalization ability of our method in comparison with recent endeavors.

## 4.3 Diagnostic Experiment

To gain deeper insights into **VIDEORFT**, we conduct a set of diagnostic experiments, as in Table 2.

Table 2: **Diagnostic experiments** for **VIDEORFT**.

| Model | Video Reasoning | | | Video Understanding | | |
|---|---|---|---|---|---|---|
| | VSI-Bench | VideoMMMU | MMVU | MVBench | TempCompass | VideoMME |
| • **Training Data** | | | | | | |
| w/o CoT Refinement | 34.5 | 48.1 | 64.8 | 58.3 | 72.4 | 52.8 |
| • **Training Paradigm** | | | | | | |
| SFT only | 31.7 | 48.5 | 60.5 | 57.0 | 68.4 | 54.1 |
| RL only | 32.1 | 47.4 | 63.5 | 59.2 | 70.8 | 51.9 |
| • **Reward Modeling** | | | | | | |
| $R = R_f + R_a$ | 33.2 | 49.1 | 66.4 | 61.1 | 72.4 | 58.5 |
| $R = R_f + R_a + R_s$ | 34.6 | 50.2 | 65.2 | 61.4 | **73.9** | 56.3 |
| • **Full Model** | | | | | | |
| **VIDEORFT** | **36.8** | **51.1** | **68.5** | **62.1** | 73.7 | **59.8** |

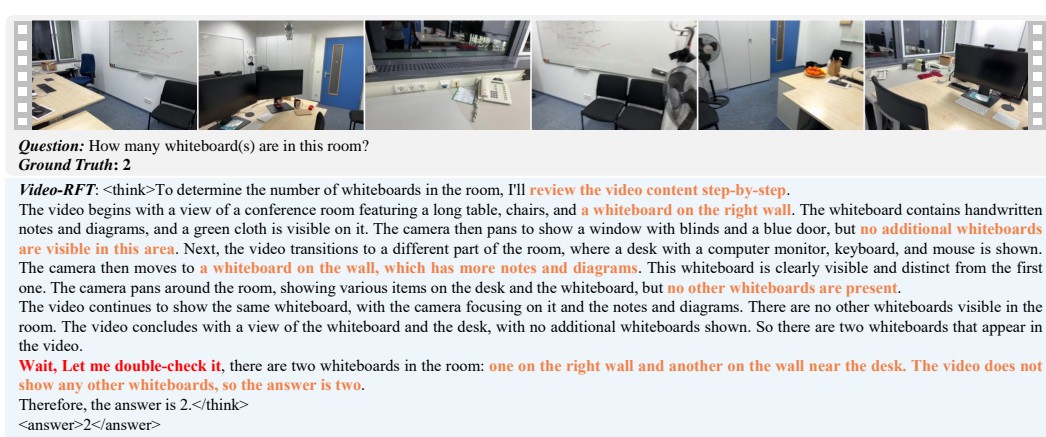

*Question:* How many whiteboard(s) are in this room?
*Ground Truth*: **2**

*Video-RFT*: <think>To determine the number of whiteboards in the room, I'll review the video content step-by-step.
The video begins with a view of a conference room featuring a long table, chairs, and a whiteboard on the right wall. The whiteboard contains handwritten notes and diagrams, and a green cloth is visible on it. The camera then pans to show a window with blinds and a blue door, but no additional whiteboards are visible in this area. Next, the video transitions to a different part of the room, where a desk with a computer monitor, keyboard, and mouse is shown. The camera then moves to a whiteboard on the wall, which has more notes and diagrams. This whiteboard is clearly visible and distinct from the first one. The camera pans around the room, showing various items on the desk and the whiteboard, but no other whiteboards are present.
The video continues to show the same whiteboard, with the camera focusing on it and the notes and diagrams. There are no other whiteboards visible in the room. The video concludes with a view of the whiteboard and the desk, with no additional whiteboards shown. So there are two whiteboards that appear in the video.
Wait, Let me double-check it, there are two whiteboards in the room: one on the right wall and another on the wall near the desk. The video does not show any other whiteboards, so the answer is two.
Therefore, the answer is 2.</think>
<answer>2</answer>

Figure 6: Illustration of reasoning traces derived from **VIDEORFT** in VSI-Bench.

**Training Data.** To assess the impact of the cross-modal refinement in Eq. 2 on data quality, we conduct SFT+RL using the data generated by Eq. 1, denoted as $\text{CoT}_v^{(0)}$, resulting in a variant, i.e., without CoT Refinement. As shown in Table 2, this consistently leads to performance drops across all six benchmarks, *i.e.*, **-2.3%** on VSI-Bench, **-3.7%** on MMVU, and **-7.0%** on VideoMME. This demonstrates that our cross-modal refinement effectively mitigates errors and hallucinations in the initial CoTs, thereby enhancing data quality and ultimately improving model performance.

**Training Paradigm.** To validate the effectiveness of the RFT training paradigm in our approach, we build two baseline approaches: *i.e.*, *SFT only* and *RL only*. The former trains the model solely with supervised fine-tuning on VideoRFT-CoT-102K, while the latter, also known as the "zero" model in DeepSeek-R1, relies exclusively on RL without prior SFT. As seen from Table 2, the *RL only* surpasses the *SFT only* counterpart on four out of six datasets, indicating the capability of RL in stimulating more generalized reasoning capabilities. When combining both stages as in RFT, our full model **VIDEORFT** achieves the best results, substantially outperforming the two baselines across all datasets. This highlights the complementary strengths of SFT for stable initialization and RL for reasoning enhancement in tackling video reasoning.

**Reward Modeling.** We further examine the effect of the reward defined in Eq. 6. Specifically, we compare two ablated variants: the first uses only the Format Reward and Accuracy Reward ($R = R_f + R_a$), while the second incorporates all three rewards directly ($R = R_f + R_a + R_s$). As seen from Table 2, adding the semantic-consistency reward $R_s$ consistently improves performance over the first variant, validating its effectiveness. Finally, our full reward formulation, which conditionally activates $R_s$ via a gating mechanism (i.e., the indicator function $\mathbb{1}[R_a > 0]$), achieves the best overall results. Notably, it brings substantial gains especially for video reasoning benchmarks, *i.e.*, **+2.2%** on VSI-Bench, **+3.3%** on MMVU, and **+3.5%** on VideoMME.

**Aha Moment in VIDEORFT.** Fig. 6 exhibits an Aha Moment in **VIDEORFT**, where it behaves in a human-like manner by pausing to double-check its inference before finalizing the answer, as seen in the phrase "Wait, let me double-check it". Such behavior suggests that the model is not simply recalling learned patterns, but is instead engaging in internal feedback loops to re-evaluate evidence and refine its inference.

## 5 Related Work

### 5.1 Multimodal Reasoning in MLLMs

Enabling reasoning in MLLMs has become a central objective in recent research [22, 35, 52, 56]. In the image domain, early works such as MMCoT [52] and DDCoT [56] disentangle perception and reasoning by treating visual understanding as input prompts for subsequent inference. In the video domain, VoT [8] and STEP [30] decompose video reasoning into predefined stages, employing template-based prompting to facilitate multi-step inference. DoraemonGPT [45] models video understanding through symbolic memory and external tool sequences, yet still follows a modular reasoning paradigm. While these methods offer structured supervision, their rigid designs often limit generalization across diverse temporal and causal scenarios. Recently, rule-based RL has emerged as a promising paradigm for promoting multimodal reasoning in MLLMs. Pioneering efforts such as Visual-RFT [26], R1-VL [49], and Reason-RFT [37] directly adapt rule-based RL to image perception tasks. Follow-ups like Vision-R1 [15] and R1-OneVision [44] further demonstrate its effectiveness in enabling CoT reasoning on images. Concurrently, this paradigm has also been explored for video understanding [9, 20, 51]. Despite encouraging progress, these methods face a fundamental bottleneck: the lack of large-scale, high-quality video CoT datasets, which limits the full potential of RFT in the video domain. Our work addresses this gap by proposing a scalable and cognitively inspired pipeline to automatically mine high-quality CoT annotations for videos. Beyond this, we introduce a novel reward modeling strategy based on cross-modal semantic consistency, which explicitly guides MLLMs to generate visually grounded reasoning traces, and proves to be highly effective in improving model performance.

### 5.2 Multimodal CoT Dataset Construction

CoT has proven effective for enhancing the reasoning capabilities of LLMs by encouraging step-by-step reasoning [41, 53]. Constructing high-quality CoT data in multimodal settings, particularly for video reasoning, remains a major challenge due to the temporal complexity and visual ambiguity of video data [12, 33, 40]. Recent works have explored CoT construction in both image and video domains. LLaVA-CoT [42], Vision-R1 [15], and R1-OneVision [44] simply convert visual inputs to textual descriptions before reasoning. This often leads to hallucinations and weak semantic alignment. Video-R1 [9] adopts a simplistic prompting strategy that encourages MLLMs to generate CoT by inserting "let me think", "wait", *etc.*into responses. However, such CoTs merely mimic the surface form of human thinking without engaging in genuine reasoning. VideoEspresso [12] generates CoT data by prompting GPT-4o with a small set of selected key frames. Due to the sparse visual context and reliance on a text-only model, the generated CoTs often lack grounding in the actual video content and are prone to hallucinations. In contrast, our CoT data combines the reasoning abilities of reasoning LLMs and the multimodal abilities of MLLMs, ensuring the reasoning depth and visual grounding of CoT data. Moreover, we use cognition-inspired prompts to enable the reasoning model to generate CoT data that is more in line with human cognition.

## 6 Conclusion

In this work, we introduce **VIDEORFT**, a novel approach for incentivizing cognitive video reasoning capabilities in MLLMs through reinforced fine-tuning. To accomplish this, we propose a cross-modal pipeline that generates high-quality cognitive video CoT data simulating human reasoning processes, resulting in two large-scale datasets: VideoRFT-CoT-102K and VideoRFT-RL-310K. Furthermore, to strengthen the RL phase, we develop semantic-consistency guided reward to explicitly encourage the alignment between reasoning traces and visual evidence. Extensive experiments across six benchmarks demonstrate that **VIDEORFT** consistently surpasses a variety of advanced MLLMs. We expect this work to lay a foundation for future efforts in RFT-based video reasoning.

## Acknowledgments

This work is supported by the NSFC (Grant Nos. 62576035, 62225203, 62532007), the National Key R&D Program of China (Grant No. 2022YFB2702100), Beijing Natural Science Foundation (L252036), CAAI-Lenovo Blue Sky Research Fund, the Beijing Municipal Science and Technology Commission and Zhongguancun Science Park Management Committee (Z231100007423003).

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

# A   Additional Experiments

## A.1   Effect of Model Scaling on Reasoning Ability

Table 3: **Performance Comparison with Small-scale Models**

| Model | Video Reasoning | | | Video Understanding | | |
|---|---|---|---|---|---|---|
| | VSI. | VideoMMMU | MMVU | MV. | TempC. | VideoMME |
| TinyLLaVA-Video-R1-3B | - | - | 46.9 | - | 49.5 | 46.6 |
| **VIDEORFT-3B** | 32.5 | 41.1 | 55.1 | 59.5 | 61.0 | 45.4 |
| **VIDEORFT-7B** | 36.8 | 51.1 | 68.5 | 62.1 | 73.7 | 59.8 |

To further assess the scalability and robustness of our method under limited computational resources, we train an additional 3B variant of our model. While its performance is naturally lower than the original 7B version, the 3B model still performs competitively across benchmarks. Crucially, when compared to TinyLLaVA-Video-R1-3B, a contemporary 3B model specifically optimized for lightweight deployment, our 3B variant outperforms it, with gains of **+8.2%** on MMVU and **+11.5%** on TempCompass. This significant margin confirms the competitiveness of our approach even in small-scale settings, and underscores its strong generalization and reasoning abilities under resource-constrained conditions.

## A.2   Hyperparameter Analysis

Table 4: **Hyperparameter experiments** for **VIDEORFT**.

| Hyperparameter | Video Reasoning | | | Video Understanding | | |
|---|---|---|---|---|---|---|
| | VSI-Bench | VideoMMMU | MMVU | MVBench | TempCompass | VideoMME |
| $w = 1$ | 34.2 | 49.2 | 67.9 | **62.6** | 73.5 | **61.1** |
| $w = 2$ | **36.8** | **51.1** | **68.5** | 62.1 | **73.7** | 59.8 |
| $w = 3$ | 35.6 | 50.1 | 67.9 | 60.8 | 72.3 | 60.4 |
| $w = 4$ | 35.4 | 49.7 | 67.6 | 62.0 | 73.1 | 59.1 |

To evaluate the effect of the scaling factor $w$ in the semantic reward, we conducted a hyperparameter study across multiple benchmarks. As shown in Table 4, the performance varies with different values of $w$, with $w=2$ consistently achieving the best or near-best results across all benchmarks. This suggests that $w=2$ provides a favorable balance between semantic alignment and factual precision. In comparison, both smaller ($w=1$) and larger ($w=3$ or $w=4$) values lead to a slight decline in performance, implying that insufficient or excessive emphasis on semantic similarity may hinder the overall reasoning and understanding capabilities. These findings highlight that while performance is relatively stable across a range of scaling values, setting $w=2$ consistently yields optimal results, demonstrating the effectiveness of moderate semantic weighting.

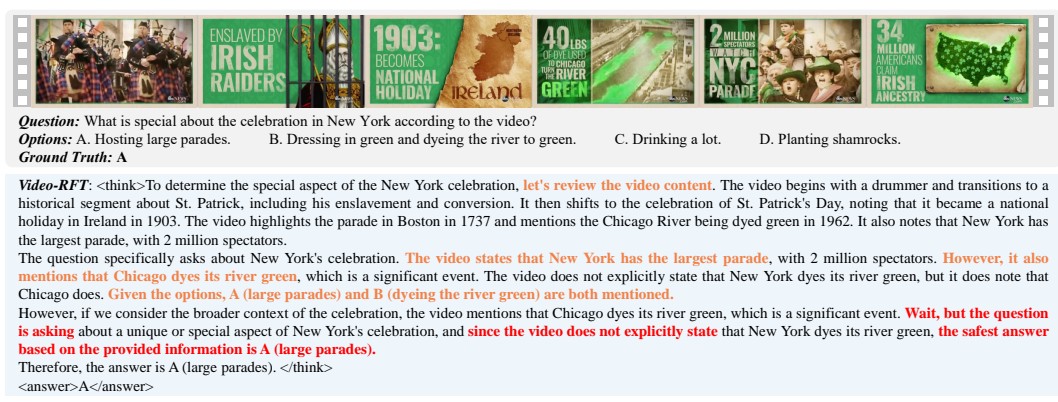

Figure 7: Illustration of reasoning traces derived from **VIDEORFT** in VideoMME.

### A.3 Aha Moment in VIDEORFT

Figure 7 illustrates additional instances of the "Aha Moment" behavior observed in **VIDEORFT**. Prior to reaching a final decision, the model exhibits a human-like tendency to pause and scrutinize its intermediate reasoning steps. This reflective process, indicative of deliberative reasoning, is marked in red within the figure.

## B Limitations and Future Directions

We consider this work a strong foundation for advancing video reasoning research using MLLMs. Several avenues warrant further investigation in future studies:

- **Handling of Challenging Video Scenarios.** While our model demonstrates strong performance across various benchmark tasks, its effectiveness may be affected under complex conditions such as rapid motion or severe visual occlusion. Incorporating finer-grained visual signals and higher frame-rate sampling may help mitigate these challenges, enabling more accurate and robust visual representations.

- **CoT Data Reliance.** The quality of the generated CoT annotations is closely tied to the capabilities of the underlying reasoning language model (e.g., DeepSeek-R1). Future work could explore leveraging more advanced reasoning models to further enhance the quality of CoT data, potentially leading to improved performance of **VIDEORFT**.

## C Potential Social Impacts

- **Positive Impacts.** Enhanced video understanding enabled by **VIDEORFT** can benefit various applications. In education, it facilitates the development of intelligent tutoring systems using video content. In security, it improves the efficiency and accuracy of surveillance video analysis. Moreover, it supports content moderation by aiding in the detection and filtering of inappropriate material.

- **Negative Impacts.** There exists a risk of misuse. Misinterpretation of video content due to over-reliance on automated analysis could lead to the spread of misinformation. Additionally, in surveillance scenarios, the deployment of such systems may raise concerns regarding privacy and ethical use.

## D Detailed Prompt Used in CoT Generation

Here, we provide detailed prompts for each step of the **VIDEORFT** CoT generation process.

### D.1 Video CoT Generation

▶ **Structured Video Representation:** For all videos, we sample them at 1FPS and input them into GPT-4o-mini, and generate a structured representation according to the prompt $P_{\text{rep}}$:

---

**Structured Video Representation**

▶ **System Prompt:**
You are a video analysis assistant designed to produce rich, analytical per-frame captions from video inputs.

▶ **User Prompt:**
`<Input Video>` 🎬 `</Input Video>`

**Task:**

1. **Overall Video Caption:**
   - `video_caption`: A concise 1-2 sentence (20-30 words) summary capturing the main theme or action of the video.

---

2. **Per-Frame Metadata:**
   Uniformly sample frames and wrap them into a JSON list. Each element must include:
   - **timestamp**: `"HH:MM:SS"`, aligned with the sampling interval.
   - **caption**: 2-3 sentences (30-50 words) describing the scene in detail.
   - **key_elements**: an object with fields:
     – **objects**: list of detected objects/entities (strings)
     – **actions**: list of ongoing actions or movements (strings)
     – **scene**: *e.g.*"kitchen", "urban street"
     – **notable_features**: list of distinctive colors, textures, or patterns (strings)
     – **spatial_relations**: list of spatial relationships (*e.g.*"cup on table", "person left of car")
     – **human_attributes**: object or `null`. If present, include:
       * **gender**: "male", "female", "unknown"
       * **clothing**: brief description
       * **posture**: "standing", "sitting"
     – **potential_interactions**: list of possible interactions (strings)

**General Instructions:**

- **Temporal Consistency:** Reference continuing actions from the previous frame and highlight any changes.

- **Uncertainty:** If confidence<0.6 or object visibility<50%, append `[Uncertain]` to the caption.

- **Implied Actions:** Describe preparatory movements (*e.g.*"hand reaching toward door handle" vs. "holding door handle").

- **Output Requirements:** Wrap all per-frame objects into a single JSON list matching the number of sampled frames.

▶ **Cognitively Inspired CoT Generation:** We invoke DeepSeek-R1 to answer the question and extracts its step-by-step reasoning outputs with the prompt $P_{\text{cog}}$ as the initial CoT.

**Cognitively Inspired CoT Generation**

▶ **System Prompt:**
You are an AI assistant helping a user answer questions about a video. When the user asks a question, you respond by imagining you are watching the video with full attention, just like a human would. Your task is to reason visually and logically about the video content to answer the user's question.

Follow this multi-step reasoning approach:

1. **Simulate Browsing the Video**: Imagine you are watching the entire video from beginning to end. Build a general sense of what is happening.

2. **Understand the Question**: Reflect on what the user is asking. Think carefully about what kind of answer is needed (*e.g.*, a fact, a reason, a comparison).

3. **Localize Relevant Moments**: Consider which parts of the video are most related to the question. Focus on those segments in your mental replay.

4. **Visual Reasoning**: Describe what you "see" in those segments using natural visual language (*e.g.*, "The video shows...", "In the second half of the video..."). Analyze and interpret the visual content to build your answer.

5. **Answer Thoughtfully**: Provide a clear and direct answer. Ensure your reasoning is consistent with the visual events you described.

Guidelines for Responses:

The `<Corresponding Answer Format Template>` in the prompt is dynamically selected from the following templates based on the question type:

Table 5: Answer Format Templates for Different Question Types

| Question Type | Template |
|---|---|
| Multiple Choice | Please provide only the single option letter (e.g., A, B, C, D, etc.) within the `<answer> </answer>` tags. |
| Numerical | Please provide the numerical value within the `<answer> </answer>` tags. |
| OCR | Please transcribe text from the image/video clearly and provide your text answer within the `<answer> </answer>` tags. |
| Free-form | Please provide your text answer within the `<answer> </answer>` tags. |
| Regression | Please provide the numerical value within the `<answer> </answer>` tags. |

▶ **Cross-modal CoT Refinement:** We employ a cross-modal refinement process to ensure the CoT aligns with the video content, using the prompt $P_{\text{cross}}$:

## D.2 Image CoT Generation

In addition to the main video data in the VideoRFT-CoT-102K, we have also designed specific prompts specifically for image data, and used the similarly CoT generation process to generate CoT.

▶ **Structured Image Representation:** Due to the differences among different image datasets (the task focuses of the datasets are different), we designed different prompts for each image dataset in structured image representation phase.

The following prompts are used for different image datasets, all sharing the above system prompt:

**General Image Representation (A-OKVQA, ShareGPT4V)**

2. **Image Metadata**: an object containing:
   - `objects`: list of objects with id, type, color, size, and bbox
   - `text`: list of text elements with content and bbox
   - `scene_context`: environment and activity
   - `relations`: list of subject-predicate-object relations

**CLEVR Image Representation**

1. **Overall Image Caption**: a brief summary of number of objects and overall layout.

2. **Image Metadata**:
   - `objects`: list of objects with id, shape, color, size, material, and coordinates
   - `spatial_relations`: list of spatial relations between objects

## STEM Image Representation (Geometry3K, UniGeo, AI2D)

▶ **User Prompt:**

`<Input Image>`  `</Input Image>`
You are given a line-drawing or diagram. Output **only** valid JSON with:

1. **Overall Image Caption**: a concise paragraph describing the diagram's purpose.
2. **Image Metadata**:
   - `primitives`: list of geometric primitives with type, label, and bbox
   - `annotations`: list of relations between primitives
   - `measurements`: list of measurements with primitive IDs and values

## OCR Image Representation (TextVQA, HME100k)

▶ **User Prompt:**

`<Input Image>`  `</Input Image>`
You are given an image containing printed or handwritten text. Output **only** valid JSON with:

1. **Overall Image Caption**: one sentence summarizing the text context.
2. **Image Metadata**:
   - `text_items`: list of text elements with content, bbox, and style

## Science Image Representation (ScienceQA, PMC-VQA, ArxivQA)

▶ **User Prompt:**

`<Input Image>`  `</Input Image>`
You are given a multi-panel scientific figure. Output **only** valid JSON with:

1. **Overall Image Caption**: a paragraph overviewing the figure's subject.
2. **Image Metadata**:
   - `panels`: list of panels with elements and process arrows

## Chart Image Representation (DVQA, PlotQA, FigureQA)

▶ **User Prompt:**

`<Input Image>`  `</Input Image>`
You are given a chart image. Output **only** valid JSON with:

1. **Overall Image Caption**: a 1-2 sentence summary of chart type and key trend.
2. **Image Metadata**:
   - `chart_type`: type of chart
   - `axes`: list of axis information
   - `series`: list of data series
   - `legend`: list of legend entries

## Math Image Representation (Multimath-300K, TQA)

▶ **User Prompt:**

`<Input Image>`  `</Input Image>`
You are given a textbook problem image. Output **only** valid JSON with:

1. **Overall Image Caption**: a summary of the problem context.
2. **Image Metadata**:
   - `equations`: list of equations with LaTeX and bbox
   - `diagram_parts`: list of diagram elements
   - `givens`: list of given values

**Spatial Image Representation (OpenSpaces, Spacellava)**

▶ **User Prompt:**
`<Input Image>`  `</Input Image>`
You are given an indoor scene or floorplan image. Output **only** valid JSON with:

1. **Overall Image Caption**: a 2-3 sentence narrative of the space.
2. **Image Metadata**:
   - `rooms_or_sections`: list of room information
   - `furniture`: list of furniture items with position and orientation
   - `annotations`: list of structural elements

▶ **Cognitively Inspired CoT Generation:**  We invoke DeepSeek-R1 to answer the question and extracts its step-by-step reasoning outputs with the prompt $P_{cog}$ as the initial CoT.

**Cognitively Inspired CoT Generation for Images**

▶ **System Prompt:**
You are an AI assistant helping a user answer questions about an image. When the user asks a question, you respond by imagining you are looking at the image with full attention, just like a human would. Your task is to reason visually and logically about the image content to answer the user's question.

Follow this multi-step reasoning approach:

1. **Simulate Visual Perception**: Imagine you are looking at the entire image carefully. Build a general understanding of what is shown.
2. **Understand the Question**: Reflect on what the user is asking. Think carefully about what kind of answer is needed (*e.g.*, a fact, a reason, a comparison).
3. **Identify Relevant Elements**: Consider which parts of the image are most related to the question. Focus on those elements in your mental analysis.
4. **Visual Reasoning**: Describe what you "see" using natural visual language (*e.g.*, "The image shows. . . ", "In the upper part of the image. . . "). Analyze and interpret the visual content to build your answer.
5. **Answer Thoughtfully**: Provide a clear and direct answer. Ensure your reasoning is consistent with the visual elements you described.

Guidelines for Responses:

- Don't expose in the output that you are answering based on text information. Use statements imitating looking at an image to answer.
- Don't directly refer to any textual metadata such as "captions", "description", "metadata", etc. If you need to mention them, use "visual evidence" instead (*e.g.*, "the image shows. . . ").
- It's okay to double-check or question yourself during the thought process — reflect naturally as a human would.

## Cognitively Inspired CoT Generation for Images — Continued

- Refer to locations in the image using expressions like: "in the center", "at the top", "on the left side", or "in the background".

▶ **User Prompt:**
Image content: `<Overall Image Caption>`
Image metadata: `<Image Metadata>`
Question: `<Question>`
Please think about this question as if you were a human pondering deeply. It's encouraged to include self-reflection or verification in the reasoning process.
`<Corresponding Answer Format Template>`

▶ **Cross-modal CoT Refinement:** We employ a cross-modal refinement process to ensure the CoT aligns with the image content, using the prompt $P_{\text{cross}}$:

## Cross-modal CoT Refinement for Images

▶ **System Prompt:**
You are a multimodal reasoning expert. Your task is to revise hallucinations and errors in the chain-of-thought (CoT) based on the provided image. Do not significantly alter the original CoT logic or content, and ensure the final conclusion remains the same.

**Your task:**

1. Carefully examine the image, the question, and the CoT.

2. Identify only the reasoning steps that directly conflict with what is shown in the image:
   - Replace all references to textual cues (such as "title", "bbox", "label") with direct visual observations from the image.
   - Use visual phrases such as "The image shows...", "I can see...", or "From the visual layout..." instead of text-based observations.
   - Use broader spatial descriptions like "on the left", "in the background", or "in the center", instead of specific coordinates or labeled boxes.
   - Do not rewrite or paraphrase steps that are already visually accurate or consistent with the image.
   - Only replace or correct parts that visually contradict what is shown.

3. Ensure the rest of the CoT (which is either correct or visually consistent) stays faithful to the original meaning. Only revise incorrect or hallucinated steps based on visual evidence.

▶ **User Prompt:**

`<Input Image>`  `</Input Image>`
Question: `<Question>`
Original CoT: `<Original CoT>`

**Output format:**
Strictly follow this format. Return only the revised CoT and no additional explanation:
`<think>[Revised CoT]</think>`

