# OpenReview forum: "VideoRFT: Incentivizing Video Reasoning Capability in MLLMs via Reinforced Fine-Tuning"
_NeurIPS.cc/2025/Conference — NeurIPS 2025 poster_

### Official Review · Reviewer_9Vu5 · 2025-06-27

**Clarity:** 3
**Significance:** 2
**Originality:** 2
**Rating:** 4
**Confidence:** 4

**Summary:**

The paper is one of the recent efforts to expand reasoning capabilities to a multi-modal large language model (MLLM), specifically for videos. For this task, one of the main hurdles is the lack of video-text datasets with step-by-step visual and text explanations. This paper discusses that such datasets are the first entry point to train MLLM with reasoning capabilities. Thus, this paper introduces an automated data pipeline for creating such a video reasoning dataset. Their proposed pipeline consists of two stages:(a) using an existing MLLM to generate a structured textual representation of the video, including captions, timestamps, etc. Then, using an existing reasoning LLM, specifically the DeepSeek model, to transform this structured text into step-by-step reasoning outputs. (b) In the second stage, another MLLM, Qwen 2.5-VL, is used to filter the generated reasoning by checking whether the reasoning traces are well-aligned with the visual content of the video. This dataset was then used to train the MLLM model, 7 B Qwen 2.5-VL. The training process follows a two-stage approach: supervised fine-tuning using the VideoRFT-CoT-102K dataset, followed by reinforcement learning via Group-Relative Policy Optimization (GRPO) using the VideoRFT-RL-310K dataset. During the reinforcement stage, the reward function combines accuracy, output formatting, and semantic consistency. The model is penalized when its reasoning trace, the portion of the response that describes video content, is misaligned with the actual video content, as measured by the SigLIP model. This penalty is applied only when the model's final answer is correct, based on the insight that even well-grounded reasoning is useless if it leads to an incorrect conclusion/answer. Their experimental results show that such reward formulation leads to performance improvements across multiple benchmarks.

**Questions:**

Suppose we take the structured text representation of a video, convert it into chain-of-thought (CoT) reasoning, apply your proposed filtering step, and then feed both the original video and the reasoning-based description to a pre-trained and frozen Qwen2.5-VL model. How would its performance compare to that of the fine-tuned version? This would help clarify whether additional fine-tuning is truly necessary or whether the reasoning improvements can come from better prompting and data conditioning.

Assuming fine-tuning is still helpful, what tradeoffs does it introduce in terms of generalization? Specifically, how does your fine-tuned model perform on other benchmarks not included in Table 1, such as EgoSchema, LongVideoBench, VideoMME with subtitles, or MMVU when not restricted to multiple-choice questions? A comparison here would help clarify whether the fine-tuning narrows or broadens the model’s capabilities.

Also, your model is trained using inputs of 128 × 28 × 28 resolution. In Table 1, are the baseline models trained under comparable input settings? If not, how is the comparison fair?

**Ethical Concerns:**

["NO or VERY MINOR ethics concerns only"]

**Final Justification:**

The rebuttal addressed my concerns, particularly regarding the need for fine-tuning compared to prompting and the generalization ability of the fine-tuned model. I have therefore improved my rating.

**Limitations:**

yes

**Paper Formatting Concerns:**

Did not notice formatting issues.

**Quality:**

3

**Strengths And Weaknesses:**

The paper is well-written and organized. The generated data pipeline is clearly described and reproducible, and so are the training details. The results are favorably better than baseline Qwen 2.5-VL 7B and similar approaches, such as Video-R1 on the selected six benchmarks/datasets.

The originality of this work mainly lies in fine-grained reward modifications by invoking semantic consistency only if the model is accurate during training.

The proposed automated data pipeline is also not that novel, except for prompting LLM with five sub-prompts mimicking human cognition.

In my view, these datasets and training MLLMs on them only truly matter if we can first establish that baseline models genuinely lack reasoning ability, and that further fine-tuning does not degrade their generalization. Please see my questions below.

---

> ### Author Rebuttal · Authors · 2025-07-31
>
> We thank the reviewer for the insights and suggestions. We address your comments below, and start with the following question:
>
> > Q1. The proposed automated data pipeline isn't that novel, except for prompting LLM with five sub-prompts mimicking human cognition.
>
> Thanks for your comments. As noted in your **Summary**,  the scarcity of CoT datasets has been a major bottleneck in video reasoning. Thus, we believe that our work,  constructing such a dataset and validating its value, is itself a timely and valuable contribution to the community.
>
> Regarding the novelty of our pipeline, we'd like to clarify its distinction from prior work. Current CoT generation pipelines typically fall into two categories: 1) Blind LLM pipelines (e.g., [ref1-2]) that rely solely on reasoning-centric LLMs (e.g., DeepSeek-R1) prompted with textual video descriptions. These methods often suffer from hallucination due to the lack of visual grounding. 2) MLLM-based approaches (e.g., Video-R1 [ref3]) that use vision-language models (e.g., Qwen-VL) to simulate reasoning via surface-level cues (e.g., "let me think…"). However, these models lack the deep reasoning capabilities needed for complex multi-step inference.
>
> In sharp contrast, our method introduces two major innovations:
>
> First, we propose **a cognitively inspired five-stage CoT reasoning structure** that goes beyond the typical approaches. Rooted in cognitive science, our method decomposes reasoning into five stages: *observation, task understanding, visual fosuc, reasoning, and reflection*. This design mirrors human reasoning processes and enables a structured transition from visual grounding to abstract thought, yielding more interpretable and accurate CoTs.
>
> Second, our pipeline establishes a collaborative framework between complementary model types. We pair the abstract reasoning power of blind LLMs (e.g., DeepSeek-R1) with the visual grounding capabilities of strong MLLMs (e.g., Qwen2.5-VL). This multi-expert, multi-stage generation and refinement process ensures that CoTs are both logically sound and visually grounded. This is rarely explored in existing video reasoning studies. Thanks.
>
>
>
>
> Next, we address the following comments:
> > Q2.1 In my view, these datasets and training MLLMs on them only truly matter if we can first establish that baseline models genuinely lack reasoning ability.
>
> > Q2.2 Suppose we take the structured text representation of a video, convert it into chain-of-thought (CoT) reasoning, apply your proposed filtering step, and then feed both the original video and the reasoning-based description to a pre-trained and frozen Qwen2.5-VL model. How would its performance compare to that of the fine-tuned version? This would help clarify whether additional fine-tuning is truly necessary or whether the reasoning improvements can come from better prompting and data conditioning.
>
> Following your suggestion, we conduct additional experiments to examine whether improved prompting alone can effectively elicit the reasoning capabilities of baseline models. The conclusion is: **better prompting remains insufficient for this purpose**.
>
> **Experimental Setup:** First of all, we note that (1) the suggested pipeline, while potentially informative, is computationally expensive and impractical for real-world deployment; (2) more importantly, the reasoning-based description (i.e., CoTs) from such a pipeline would implicitly include the final answers, making the evaluation less meaningful. Therefore, we instead explore two alternative prompting strategies, and feed the video, question, and prompts to the baseline to derive answers:
>
> * **Plain Prompting:** We design a plain prompt to encourage the baseline to explicitly reason within a `<think>...</think>` block before answering, and  use reflective phrases such as "wait", "let me think", etc. The plain prompt is:
> > {Question}
> > Please think about this question as if you were pondering deeply.  Engage in an internal dialogue using expressions such as 'let me think', 'wait', 'oh, I see','let's break it down', etc.  It's encouraged to include self-reflection or verification in the reasoning process. Provide your detailed reasoning between the `<think>` and `</think>` tags, and then give your final answer between the `<answer>` and `</answer>` tags."
>
> * **Cognition-Aware Prompting:** Further, we extend the plain prompt into a more complex, cognition-aware prompt by explicitly guiding the baseline to mimic the 5-step human reasoning trajectory, as presented in **Sec. 2.2**. We also guide the baseline to align its reasoning with the actual video input. The key part of the prompt is:
>
> > Follow this multi-step reasoning approach:
> > 1. Browsing the Video: You watch the entire video from beginning to end. Build a general sense of what is happening.
> > 2. Understand the Question: Reflect on what the user is asking. Think carefully about what kind of answer is needed.
> > 3. Localize Relevant Moments: Consider which parts of the video are most related to the question. Focus on those segments in your mental replay.
> > 4. Visual Reasoning: Analyze and interpret the visual content and perform in-depth reasoning to build your answer.
> > 5. Answer Thoughtfully: Reflect your reasoning and provide a clear and direct answer. Ensure your reasoning is consistent with the visual events you described.
> >
> >    [Plain Prompt]
>
>
>
> **Results are:**
> |Model|VSI-Bench|VideoMMMU|MMVU|MVBench|TempCompass|VideoMME|
> |:-|:-:|:-:|:-:|:-:|:-:|:-:|
> |Plain Prompting|31.8|47.4|61.3|59.4|69.2|52.8|
> |Cognition-Aware Prompting|34.8|49.4|58.4|58.2|72.7|55.5|
> |VideoRFT|36.8|51.1|68.5|62.1|73.7|59.8|
>
> As seen, both prompting strategies yield significantly lower performance than our VideoRFT, especially on datasets such as VSI-Bench, VideoMMMU, and MMVU, which emphasize complicated reasoning. This safely validate that **additional fine-tuning is truly necessary**.
>
>
>
> Further, you ask some follow-up questions on model generalization:
> > Q3. Assuming fine-tuning is still helpful, what tradeoffs does it introduce in terms of generalization? Specifically, how does your fine-tuned model perform on other benchmarks not included in Table 1, such as EgoSchema, LongVideoBench, VideoMME with subtitles, or MMVU when not restricted to multiple-choice questions?
>
> Sorry for the confusion.  In the current experiments, we have extensively validated that our model shows promising performance across a variety of tasks, including spatial reasoning, scientific QA, video understanding, and temporal reasoning. This already indicates compelling generalization performance.
>
> To fully address your concern, we follow the suggestions and verify our model on a set of new and more diverse datasets. These include **LongVideo Bench, VideoMME (with subtitles), IntentQA, Next-QA, and Causal-VidQA**. Results are:
>
> |Model|LongVideo Bench|VideoMME (with subtitles)|IntentQA|Next-QA|Causal-VidQA|
> |:-|:-:|:-:|:-:|:-:|:-:|
> |Qwen2.5-VL|36.0|55.5|86.3|75.6|70.2|67.0|
> |Video-R1|50.2|62.1|88.2|78.3|61.7|56.0|
> |VideoRFT|54.7|63.3|92.7|80.1|71.6|67.0|
>
>
> The significant improvements of VideoRFT against Qwen2.5-VL  confirms that **fine-tuning substantially broadens the baseline's capabilities in video reasoning**. Moreover, the results also show that our model  exhibits stronger generalization compared to Video-R1.
>
>
>
> > Q4. Your model is trained using inputs of 128 × 28 × 28 resolution. In Table 1, are the baseline models trained under comparable input settings? If not, how is the comparison fair?
>
> Apologize for the confusion. We clarify that there are two commonly used input resolution settings in the competitors. On one hand, same to us, models such as Video-R1, VILA, and LLaMA-VID use 128 × 28 × 28 resolution (equivalent to 224 × 224). On the other hand, some baselines, including TinyLLaVA-Video-R1, mPLUG-Owl3, LLaVA-OneVision, and VideoLLaMA2, use a higher resolution of 384 × 384.
>
> Therefore, the comparison is fair and, if anything, conservative, as some competing methods benefit from larger input resolutions than ours.
>
> [ref1] Vision-R1: Incentivizing Reasoning Capability in Multimodal Large Language Models, Arxiv 2025
>
> [ref2] R1-Onevision: Advancing Generalized Multimodal Reasoning through Cross-Modal Formalization, Arxiv 2025
>
> [ref3] Video-R1: Reinforcing Video Reasoning in MLLMs, Arxiv 2025

---

> > ### Comment · Reviewer_9Vu5 · 2025-08-09
> > **Thanks for detailed rebuttal**
> >
> > Thanks, the rebuttal addressed my concerns. I will consider this in the final rating.

---

> ### Author Response · Authors · 2025-08-05
>
> Dear Reviewer 9Vu5,
>
> Thanks once again for your valuable comments on our submission. As the discussion phase is approaching its end, we would like to kindly confirm whether we have sufficiently addressed all of your concerns (or at least part of them), and if there are any additional ones. We sincerely hope that we will be able to use the remaining time to engage in an open dialogue with domain experts to enhance the quality of our work.
>
> Thanks, Authors

---

> ### Author Response · Authors · 2025-08-08
>
> Dear Reviewer 9Vu5,
>
> As the discussion phase is closing soon, we would like to kindly follow up once more to check whether any remaining concerns need clarification from our side. We truly appreciate your time and feedback, and would be grateful for any final comments or confirmations you may have.
>
> Thanks, Authors

---

> ### Author Response · Authors · 2025-08-09
>
> Dear Reviewer 9Vu5,
>
> As the discussion phase is closing soon, we would like to kindly follow up once more to check whether any remaining concerns need clarification from our side. We truly appreciate your time and feedback, and would be grateful for any final comments or confirmations you may have.
>
> Thanks, Authors

---

### Official Review · Reviewer_3mwX · 2025-06-30

**Clarity:** 2
**Significance:** 2
**Originality:** 3
**Rating:** 3
**Confidence:** 4

**Summary:**

The paper introduces VIDEORFT, a novel framework to enhance video reasoning capabilities in MLLMs through Reinforced Fine-Tuning (RFT). The framework extends traditional RFT into the video domain by addressing two critical challenges: (1) the scarcity of large-scale, high-quality CoT datasets for video reasoning, and (2) the lack of mechanisms to ensure alignment between textual reasoning and visual evidence. VIDEORFT's contributions include: A scalable, cognition-inspired pipeline for generating high-quality CoT data for video reasoning. This results in two datasets: VideoRFT-CoT-102K for supervised fine-tuning (SFT) and VideoRFT-RL-310K for reinforcement learning. A novel semantic-consistency reward that promotes grounding of the reasoning process in visual evidence, reducing hallucinations and improving reasoning coherence. Experimental results demonstrating state-of-the-art performance across six video reasoning benchmarks.

**Questions:**

The leading performance of VIdeo-R1 is caused by what factors? The reward design or the data's quality?

Why didn't the author try to increase the training frames to verify the video reasoning effect under different frames? Did the author also set the inference frames to 32 when reasoning video-R1?

Will the Semantic-Consistency Reward conflict with the reasoning-specific content, such as "wait", "let me think". Does it overly constrain the model’s reasoning capabilities, especially when abstract reasoning extends beyond visual input?

**Ethical Concerns:**

["NO or VERY MINOR ethics concerns only"]

**Final Justification:**

I still stand by the rejection of this paper, for the following reasons:

1. In the rebuttal state, the author showed the unrealistic speed of Siglip during the RL training, which is extremely unprofessional.

2. This paper introduces the 'Semantic-Consistency Reward', which requires the structural reasoning process (e.g., describe first, then think, and finally give the answer), and the authors manually set the maximum length of the first stage to 64. This would potentially break the capability of base VLM. Whether in LLM or VLM's reasoning RL, we hope the model discovers the reasoning trace freely. However, this structural setting will limit this exploration; this limitation was already confirmed by many works.

3. On the most used benchmark, VideoMME, the performance only improves 0.2%; however, it costs huge RL training resources.

**Limitations:**

The authors did not include the limitation discussion in this paper.

**Paper Formatting Concerns:**

N

**Quality:**

2

**Strengths And Weaknesses:**

Strengths:

1. The semantic-consistency reward is a meaningful contribution that ensures reasoning is grounded in visual input, tackling a significant limitation in existing MLLMs.

2. The automated pipeline for generating cognitively grounded CoT data is a notable innovation. The datasets, VideoRFT-CoT-102K and VideoRFT-RL-310K, cover diverse cognitive and reasoning tasks.

3. The paper provides detailed explanations of its novel pipeline, training strategy (SFT and RL), and reward modeling.

Weakness:

1. The semantic-consistency reward relies on SigCLIP embeddings, which may add computational overhead during training. The authors did not evaluate the consumption of this side.

2. One of the main contributions of this paper is the building of data, but the author did not show the ablation study on the dataset. I suggest that the author train their Qwen2.5-VL under the same hyperparameter setting (VideoRFT and Video-R1). It will be clearer how this dataset advanced to Video-R1.

3. The authors should be clearer about the experimental details, such as what reinforcement learning training tools were used, such as VERL, Open-R1, etc., and should cite them correctly.

4. The improvement is marginal compared to Video-R1, especially on the VideoMME dataset.

---

> ### Author Rebuttal · Authors · 2025-07-31
>
> We thank the reviewer for the feedback and suggestions. We address your comments below, and start with the following question:
>
> > Q1. The semantic-consistency reward relies on SigCLIP embeddings, which may add computational overhead during training. The authors did not evaluate the consumption of this side.
>
> Thanks for pointing this out. First, we clarify that  for SigCLIP, we use the *siglip-so400m-patch14-384* version, which has **only  400M  parameters** - very lightweight compared to the **7B parameters of the full model**. Further, per your request, we quantify the overhead on a machine with  8×A800 GPUs. As shown in the table below, SigCLIP introduces only **0.3 GB** additional memory per GPU and adds **0.03 seconds** per training step, indicating that the computational cost from SigCLIP is minimal.
>
> |Setting|Mem per GPU (GB)|Time per Step (s)|
> |:-|:-:|:-:|
> |w/o $SigCLIP$|60.3|0.42|
> |w/ $SigCLIP$|60.6|0.45|
>
> The table will be added to the paper.
>
>
>
>
> Next, we address your comments on evaluation (regarding both data and model):
> > Q2.1. **Data**: One of the main contributions of this paper is the building of data, but the author did not show the ablation study on the dataset. I suggest that the author train their Qwen2.5-VL under the same hyperparameter setting (VideoRFT and Video-R1). It will be clearer how this dataset advanced to Video-R1.
>
> > Q2.2. **Model**: The improvement is marginal compared to Video-R1, especially on the VideoMME dataset.
>
>
> **More results on our data**
>
> As suggested, we train Qwen2.5-VL using data from Video-R1 and our proposed VideoRFT during the supervised finetuning stage (SFT). All configurations are kept identical except for the training data.  The results are presented in the table below. Notably, while **VideoRFT contains only 62% of the supervision data** compared to Video-R1 (102K vs. 165K), it  leads to  better overall performance. This confirms the advantage of our dataset.
>
> |Model|VSI-Bench|VideoMMMU|MMVU|MVbench|TempCompass|VideoMME|
> |:-|:-:|:-:|:-:|:-:|:-:|:-:|
> |Qwen2.5-VL + Video-R1 dataset|31.8|**47.4**|61.3|59.4|69.2|52.8|
> |Qwen2.5-VL + VideoRFT dataset|**32.7**|47.1|**63.5**|**59.9**|**69.6**|**53.0**|
>
> **Model performance against Video-R1**
>
> **First**, we clarify that our approach shows better overall performance over Video-R1. Particularly, designed for video reasoning, more pronounced improvements on reasoning-intensive benchmarks. For instance, VideoRFT surpasses Video-R1 by +1.0% on VSI-Bench and +4.7% on MMVU. **Second**, to further address your concern, we extend the evaluation to include  5 additional video reasoning datasets.  As seen,  VideoRFT consistently outperforms Video-R1 by clear margins. This further confirms the superior performance of VideoRFT. Thanks.
>
> |Model|LongVideo Bench|IntentQA|Next-QA|Causal-VidQA|DarkEventInfer-Test|
> |:-|:-:|:-:|:-:|:-:|:-:|
> |Video-R1|50.2|88.2|78.3|61.7|56.0|
> |VideoRFT|**54.7** (+4.5)|**92.7**(+4.5)|**80.1**(+1.8)|**71.6**(+9.9)|**67.0**(+11.0)|
>
> Will add the new results to the paper.
>
>
>
> >  Q3. The authors should be clearer about the experimental details, such as what reinforcement learning training tools were used, such as VERL, Open-R1, etc., and should cite them correctly.
>
> Agree. For RL training, we use the HuggingFace TRL library. The codebase is based on Open-R1. Will add proper citations in the next version. Thanks.
>
>
>
> > Q4. The leading performance of VIdeo-R1 is caused by what factors? The reward design or the data's quality?
>
> **Both**.
>
> **Reward Design**: As shown in Table 2, removing our proposed semantic-consistency reward leads to a 2.0% average accuracy drop across all benchmarks, confirming it as a key factor in leading performance.
>
> **Data Quality**: To isolate the impact of data quality, we retrain Qwen2.5-VL using either the Video-R1 dataset or our proposed VideoRFT dataset under identical training configurations. The results are shown in the above second table, even though our dataset contains only 62% of the supervision data compared to Video-R1 (102K vs. 165K), it consistently achieves better performance across six benchmarks.
>
> To further validate the contribution of our data, we construct an alternative CoT dataset by prompting DeepSeek-R1 with our structured video representations to generate CoTs. We then train a model using this CoT data and compare it with the same model trained on our own CoTs. As shown below, our CoT data leads to consistent and notable gains:
>
> |Model Variant|VSI-Bench|VideoMMMU|MMVU|MVbench|TempCompass|VideoMME|
> |-|-|-|-|-|-|-|
> |DeepSeek-R1 output CoT|30.2|45.0|61.8|58.2|66.4|49.7|
> |Our CoT|**36.8**|**51.1**|**68.5**|**62.1**|**73.7**|**59.8**|
>
> These results demonstrate that both our carefully designed reward and high-quality CoT data are essential contributors to the performance gains of VideoRFT, each providing measurable benefits individually.
>
>
>
> > Q5. Why didn't the author try to increase the training frames to verify the video reasoning effect under different frames? Did the author also set the inference frames to 32 when reasoning video-R1?
>
> Sorry for the confusion. **First**, we clarify that our choice of the setting (16 frames for train and 32 frames for inference') strictly follows the practices in recent literature [ref1–3]. This **guarantees a fair comparison**.
>
> Due to the time limit in the rebuttal stage, we quickly examine how our model performs when increasing the number of frames during inference. As seen from the table below, when increasing the frame number from 32 to 64 to 128, the model performance progressively increases. We will continue to study the impact when increase the training frames and add the results to the next version.
>
> |Frames|32|64|128|
> |:-|:-:|:-:|:-:|
> |VideoMME|59.8|62.0|64.5|
>
> **Second**, yes, we also set the inference frames to 32 for Video-R1.
>
>
>
>
> >  Q6. Will the Semantic-Consistency Reward conflict with the reasoning-specific content, such as "wait", "let me think". Does it overly constrain the model's reasoning capabilities, especially when abstract reasoning extends beyond visual input?
>
> As noted in Sec. 3.2.1, our CoTs typically consist of three stages: *question parsing, video description, and abstract reasoning*. During reward computation, we apply the semantic-consistency reward to a fixed-length span of M tokens starting from the second sentence of the CoT. We set M = 64 for all experiments. **This span usually corresponds to the video description part and does not cover the abstract reasoning**.
>
> To directly address your concern, we analyze the average token index of (i) the start of the second sentence (where the reward span begins) and (ii) the first occurrence of abstract reasoning-specific content (e.g., "wait", "let me think") as following:
>
> |Metric|VSIBench|VideoMMMU|MMVU|MVbench|TempCompass|VideoMME|
> |:-|:-:|:-:|:-:|:-:|:-:|:-:|
> |Avg. Token Index of 2nd Sentence Start (Reward Span Begins)|21.29|23.24|21.62|17.65|17.25|19.41|
> |Avg. Token Index of 1st Abstract Reasoning Phrase|95.16|136.47|114.10|125.93|93.53|124.25|
>
> As shown, the semantic-consistency reward is applied well before the onset of general or abstract reasoning. This confirms that our reward design does not interfere with the model’s abstract reasoning capabilities.
>
>
>
> > The authors did not include the limitation discussion in this paper.
>
> Sorry for confusion. We have presented the limitation analysis in Sec. C of the supplementary.
>
> [ref1] ShareGPT4Video: Improving Video Understanding and Generation with Better Captions, NeurIPS 2024
>
> [ref2] Tinyllava-Video-R1: Towards Smaller LLMs for Video Reasoning. arXiv 2025.
>
> [ref3] Video-r1: Reinforcing Video Reasoning in MLLMs, arXiv 2025

---

> > ### Comment · Reviewer_3mwX · 2025-08-05
> >
> > Thank you for your response. However, regarding the speed of SigLip and the correctness of the reward, your reply not only fails to address my concerns but also raises doubts about its professionalism. Furthermore, the training accuracy reward curve provided in Figure 6(a) shows no upward trend, which makes me question the correctness of the RL implementation. Therefore,  I maintain my rejection of this paper.
> >
> > A1: I have used SigLip many times in VLM training. Although it only has 400M parameters, its speed becomes a significant issue during training, especially for RL. When the batch size increases and multi-frame videos are involved, the processing time is heavily impacted. The 0.03s overhead you provided is unrealistic for rollouts with a batch size greater than 16 during normal training, and I seriously doubt the accuracy of the reported data.
> >
> > A2: Regarding your data cleaning process, if the FP16 DeepSeek-R1 671B model is used, it would require 2 nodes (16 GPUs) to clean the data, which incurs a very high deployment cost. If the FP8 version of the 671B model is used, at least one node with Hopper GPUs would be necessary. However, in the experimental setup section, the authors only mentioned using 8 A800 GPUs?
> >
> > A6: Your response overlooks a critical issue: reasoning and video description do not have a strict structural separation during generation, making your reliance on average token indices to claim no interference from the reward mechanism fundamentally flawed. Reasoning content (e.g., "let me think") can overlap or interweave with video description, and the fixed reward span (M=64) could inadvertently penalize early or interleaved reasoning behavior.

---

> > > ### Author Response · Authors · 2025-08-05
> > >
> > > **Regarding A1:**
> > >
> > > We appreciate your follow-up and would like to clarify a mistake in our previous runtime report.
> > >
> > > Due to a **unit conversion oversight**, we incorrectly computed the per-step time by dividing `9:09` (9 minutes 9 seconds) as if it were a decimal (i.e., 9.09 / 20), rather than correctly converting it from minutes to seconds. This caused the previously reported **0.45 s/step** to be inaccurate.
> > >
> > > Below, we provide the **actual logs** used for measuring rollout time, both with and without SigLIP:
> > >
> > > * **With SigLIP**:
> > >
> > >   ```bash
> > >   {'loss': 0.0001, 'grad_norm': 32.507215029483376, 'learning_rate': 9.99999545465254e-07, 'completion_length': 257.76788330078125, 'rewards/accuracy_reward': 0.545666515827179, 'rewards/format_reward': 1.0, 'rewards/semantic_reward': 0.2537493109703064, 'all_wrong': 0.0, 'all_correct': 0.2857142857142857, 'reward': 1.8315588235855103, 'reward_std': 0.24415504932403564, 'kl': 0.0026092529296875, 'epoch': 0.0}
> > >   0%|          | 20/44268 [09:09<356:10:56, 28.98s/it]
> > >   ```
> > > * **Without SigLIP**:
> > >
> > >   ```bash
> > >   {'loss': 0.0001, 'grad_norm': 9.449676475422246, 'learning_rate': 9.99999545465254e-07, 'completion_length': 278.01788330078125, 'rewards/accuracy_reward': 0.5878309607505798, 'rewards/format_reward': 1.0, 'all_wrong': 0.0, 'all_correct': 0.2857142857142857, 'reward': 1.6271166801452637, 'reward_std': 0.1549045592546463, 'kl': 0.002716064453125, 'epoch': 0.0}
> > >   0%|          | 20/44268 [08:46<323:15:22, 27.83s/it]
> > >   ```
> > >
> > > Correcting the units, we obtain the actual per-step times:
> > >
> > > * 9 min 9 sec = 549 sec → 549 / 20 = **27.45 s/step**
> > > * 8 min 46 sec = 526 sec → 526 / 20 = **26.30 s/step**
> > >
> > > ### Corrected Runtime Table
> > >
> > > | Setting | Time per Step (s) |
> > > | :- | :-: |
> > > | w/o $SigLIP$ | 26.30 |
> > > | w/ $SigLIP$  | 27.45 |
> > > | Overhead | **+1.15** |
> > >
> > > This shows that **SigLIP introduces only \~1.15 seconds per step**.
> > >
> > > We also emphasize again that **SigLIP is not part of training or backpropagation—it is only used for reward scoring during RL rollouts in inference mode.**
> > >
> > > Our RL is implemented directly on the R1-V codebase, which adopts a decoupled architecture where 7 GPUs are used for training (`per_device_train_batch_size = 1`), and a separate GPU handles completion generation and reward scoring. **So the batch size is 7 per step.**
> > >
> > > We hope this clarifies the earlier misunderstanding.
> > >
> > > **Regarding A2:**
> > > We clarify that **DeepSeek-R1 is accessed via API calls** during data processing. We do not perform any local deployment.
> > >
> > > **Regarding A6:**
> > > We respectfully clarify two points:
> > >
> > > First, the token index analysis was **not intended to prove perfect separation**, but to support our claim that the fixed reward span (M=64) **usually corresponds to the video description segment**, and typically does not overlap with abstract reasoning.
> > >
> > > Second, regarding structural separation: **our model is cold-started with structured CoT data** following a clear three stage format—question parsing, video description, and abstract reasoning. This structure is learned during supervised finetuning and, based on our empirical observations, is consistently preserved during RL rollouts.

---

> ### Author Response · Authors · 2025-08-05
>
> Dear Reviewer 3mwX,
>
> Thanks once again for your valuable comments on our submission. As the discussion phase is approaching its end, we would like to kindly confirm whether we have sufficiently addressed all of your concerns (or at least part of them), and if there are any additional ones. We sincerely hope that we will be able to use the remaining time to engage in an open dialogue with domain experts to enhance the quality of our work.
>
> Thanks, Authors

---

### Official Review · Reviewer_CSJu · 2025-07-01

**Clarity:** 3
**Significance:** 2
**Originality:** 3
**Rating:** 4
**Confidence:** 3

**Summary:**

This work designed a dataset generation pipeline to build two video reasoning CoT datasets VideoRFT-CoT-102K for SFT and VideoRFT-RL-310K for RL. Based on these datasets, a MLLM is finetuned with an extra reward (Semantic-Consistency Reward). The finetuned model is evaluated on several video datasets.

**Questions:**

See weakness above.

**Ethical Concerns:**

["NO or VERY MINOR ethics concerns only"]

**Final Justification:**

See my response.

**Limitations:**

The limitation of this work is not discussed in the conclusion section.

**Quality:**

2

**Strengths And Weaknesses:**

Strengths:

- The paper is easy to understand.
- The produced dataset is a contribution to the community.

Weakness:

- From the implementation, the VideoRFT samples 16 frames from each video for training, and 32 frames for testing. It looks like this method is designed for short videos. Nowadays, for complex video reasoning tasks, the long-form videos are important part to be evaluated. Considering the recent methods shown in the VideoMME leaderboard (InternVL2.5, Qwen2-VL) that uses much more frames from a video, I doubt the long-form reasoning capability from the proposed method.
- The proposed method doesn’t achieve the SoTA performance in some datasets. For example, in VideoMME leaderboard, the best open source and published 7B/8B size baseline is ViLAMP (ICML 2025) which achieves 67.5 without subtitle, however, the proposed method only get 59.8, which is around 8% gap.
- For the dataset generation, high-quality annotations are required. However, I’m not sure if the current pipeline is sufficient to produce the high-quality dataset annotation.
    - In the first step extracting the video representation, how to ensure the GPT-4o-mini produce the reliable results without hallucination?
    - When using Qwen2.5-VL-7B to refine CoT, how to ensure this VLM can generate reliable output? Although it is designed to fix the hallucination problem from DeepSeek-R1, but how to know if Qwen2.5-VL will produce more hallucination?
    - There is no quantitative analysis for the hallucination problem from dataset annotation generation.
- The MLLM is finetuned based on the data produced from other LLM/VLMs, does it mean it is just a knowledge distillation from GPT-4o-mini, DeepSeek R1 and Qwen2.5-VL-7B?

Minor Weakness:

- Figure 2 typo: A-OKVQ → A-OKVQA

I think it is an interesting work but requires more polishing. Therefore I suggest borderline reject. I’m willing to adjust my rating if my concerns are solved.

---

> ### Author Rebuttal · Authors · 2025-07-31
>
> We thank the reviewer for the valuable feedback and suggestions. We begin by addressing the following concern:
> > Q 1. From the implementation, the VideoRFT samples 16 frames from each video for training, and 32 frames for testing. It looks like this method is designed for short videos. Nowadays, for complex video reasoning tasks, the long-form videos are important part to be evaluated. Considering the recent methods shown in the VideoMME leaderboard (InternVL2.5, Qwen2-VL) that uses much more frames from a video, I doubt the long-form reasoning capability from the proposed method.
>
>
> Thanks for your thoughtful comments, but there might be a slight misunderstanding here. We clarify the following points:
>
>
> **First**, our frame sampling setup (16 frames for training, 32 for testing) is consistent with many recent works (Video-R1, TinyLLaVA-Video-R1 and [ref1]), and was adopted to ensure a fair comparison.
>
> **Second**, although our current evaluation uses a limited number of frames, our model is not inherently restricted to short videos. In fact, it makes no assumptions about input video length. As you noted, Qwen can handle very long-context inputs (i.e., 768 frames in the VideoMME leaderboard), and since our method is built upon Qwen, it is theoretically capable of handling such long-form inputs. However, increasing the number of frames during training and inference incurs significant GPU memory and computational overhead, which exceeded our compute constraints.
>
> **Third**,  we fully agree that long-form video reasoning is an increasingly important area. Nevertheless,  our current work focuses primarily on generic video reasoning, and accordingly, we follow the standard evaluation protocols used in closely related studies (e.g., [ref1–3]) to render more meaningful validation. As such, missing evaluations in the mentioned aspect should not be a major issue.  Despite this, to further address your concern, we quickly conduct two sets of new experiments to evaluate our method's capacity in handling long-form videos:
>
> **Experiment 1: Scaling up input frames on VideoMME**
>
> We evaluate how the model benefits from increased frame counts during inference. As shown below, the model performance steadily improves with more frames, suggesting that VideoRFT can effectively leverage longer video contexts.
>
> |# frames|32|64|128|
> |:-|:-:|:-:|:-:|
> |VideoRFT|59.8|62.0|64.5|
>
> **Experiment 2: Evaluation on LongVideoBench**
>
> We further evaluate our model on the LongVideoBench dataset, which is explicitly designed to test long-form video reasoning. All models are evaluated under the same condition using 32 sampled frames. As shown below, VideoRFT outperforms all baselines. This confirms the superior capability of VideoRFT in long-form video reasoning.
>
> |Model|Qwen2.5-VL|Video-R1|VideoChat-R1|**VideoRFT**|
> |:-|:-:|:-:|:-:|:-:|
> |LongVideoBench|41.8|50.2|43.1|**52.6**|
>
>
>
>
> Next, we address another follow-up concern:
> > Q 2. In VideoMME leaderboard, the best open source and published 7B/8B size baseline is ViLAMP (ICML 2025) which achieves 67.5 without subtitle, however, the proposed method only get 59.8, which is around 8% gap.
>
>
> Thanks for referring this to us. However, we respectfully note that a direct comparison between ViLAMP and our model is not entirely fair, as ViLAMP adopts a significantly more resource-intensive setup. Specifically:
> * Training data: ViLAMP uses 1.3M shot videos and 500K longer videos for training, 17.6 times larger than us (only 102K videos);
>
> * Number of frames: ViLAMP supports up to 600 frames for both training and inference, whereas we use only 16 frames for training and 32 for inference.
> * Image resolution:  ViLAMP uses higher-resolution frames (384×384) compared to our model’s 224×224.
>
> To enable a more reasonable comparison, we perform a new evaluation on the VideoMME dataset using 128 frames during inference.  p.s. this is still not strictly fair as our model was still trained on 16 frames and uses a smaller image resolution.
>
> The results are:
> ||Short|Medium|Long|
> |:-|:-:|:-:|:-:|
> |ViLAMP|75.1|64.6|53.2|
> |VideoRFT|76.3|66.1|51.1|
>
> As seen, our model outperforms ViLAMP on the Short and Medium subsets.  While our performance is slightly lower on the Long subset, we believe this is expected, as ViLAMP is specifically tailored for long-form video reasoning, whereas our approach focuses on general-purpose video reasoning without such long-form-specific optimizations.
>
>
>
> Regarding hallucination, you mentioned:
> > Q 3.1. In the first step extracting the video representation, how to ensure the GPT-4o-mini produce the reliable results without hallucination?
>
> > Q 3.2. When using Qwen2.5-VL-7B to refine CoT, how to ensure this VLM can generate reliable output? Although it is designed to fix the hallucination problem from DeepSeek-R1, but how to know if Qwen2.5-VL will produce more hallucination?
>
> > Q 3.3. There is no quantitative analysis for the hallucination problem from dataset annotation generation.
>
> Thanks for your comments. We fully acknowledge that hallucination is a well-known and ongoing challenge for large models. In our method,  we have carefully designed our data generation pipeline to minimize hallucinations throughout. Below, we address each of your specific concerns:
>
>  **Regarding Q 3.1**
>
> In the first stage of our pipeline, we adopt a **structured prompting** strategy that constrains GPT-4o-mini’s output to a strict JSON format, specifying fields such as timestamp, scene, object, and action. This structure explicitly grounds the model’s responses in observable video elements and enforces output consistency. The idea of structured prompting has been validated in prior works such as [ref4], that is, structured prompting  significantly reduces hallucination rates—for example, by 76.9% in the Wikidata list generation task. Additionally, our approach performs automatic format validation and discard ill-formed outputs.  These steps collectively help ensure that the extracted video representations from GPT-4o-mini are  properly grounded in the video with minimal hallucination.
>
> **Regarding Q 3.2**
>
> We clarify that our pipeline use the 32B (rather than 7B) version of Qwen2.5-VL here, for refining CoT annotations. While hallucination still cannot be entirely avoided, we find that grounding the CoT refinement in visual evidence is a much more important factor in improving reliability. In our pipeline, Qwen2.5-VL-32B is provided with video-grounded representations and is tasked with validating and refining prior CoTs generated by **blind LLMs** (i.e., DeepSeek-R1). This refinement strategy improves factual consistency and reduces hallucination by anchoring reasoning to explicit visual context..
>
> **Regarding Q 3.2**
>
> To quantitatively analyze hallucinations from dataset annotation, we consider both direct data inspection and its downstream impact on model performance.
>
> First, following [ref5], we randomly sample 200 CoTs from the VideoRFT-CoT dataset. GPT-4o extracts all referenced entities and actions from CoTs and checks whether they are grounded in the corresponding video/image. Potentially ungrounded items are then manually verified. Finally, we observe a small hallucination rate of **4.5%** in our final dataset. In contrast, a variant without the cross-modal CoT refinement stage showed a significantly higher hallucination rate of **33.0%**.
>
>
> Second, we evaluate the downstream impact of hallucinated CoTs by training a model with unrefined annotations. Performance drops significantly across all benchmarks:
>
> |Model Variant|VSI-Bench|VideoMMMU|MMVU|MVbench|TempCompass|VideoMME|
> |:-|:-:|:-:|:-:|:-:|:-:|:-:|
> |w/o Cross-Modal CoT Refinement|30.2|45.0|61.8|58.2|66.4|49.7|
> |Full VideoRFT|36.8|51.1|68.5|62.1|73.7|59.8|
>
> These results confirm the effectiveness of our dataset annotation pipeline in progressively reducing hallucination, ultimately producing high-quality CoTs with a low hallucination rate.
>
>
>
> > Q 4. The MLLM is finetuned based on the data produced from other LLM/VLMs, does it mean it is just a knowledge distillation from GPT-4o-mini, DeepSeek R1 and Qwen2.5-VL-7B?
>
>
> Thank you for the question. While it's true that our model is initially fine-tuned using data generated by other LLMs/VLMs, it would be reductive to characterize our approach as merely knowledge distillation from them.
>
>
> Video reasoning is a particularly challenging task due to the scarcity of high-quality, structured CoT annotations. To tackle this, we design an automated pipeline to distill knowledge from other LLM/VLMs. However, the distillation scheme goes far beyond conventional technique, and takes a multi-modal, multi-expert collaboration pipeline to integrate complementary strength from diverse sources:  the visual grounding of VLMs and the deep reasoning abilities of LLMs.
>
> Beyond this, we clarify that the main contribution of our method lie in a reinforced fine-tuning paradigm for video reasoning. As evidenced in Table 2,  the most substantial performance gains arise from the RL phase (rather than SFT).  As such, our method is not just a passive recipient of distilled knowledge, but rather an active framework for knowledge integration, adaptation, and generalization tailored specifically for complex video reasoning. Thanks.
>
>
>
> > Q 5. Minor Weakness: Figure 2 typo: A-OKVQ → A-OKVQA
>
> Will fix it. Thanks.
>
> > Q 6. The limitation of this work is not discussed in the conclusion section.
>
> The limitation has been discussed in Sec. C of the supplementary material.
>
> [ref1] ShareGPT4Video: Improving Video Understanding and Generation with Better Captions, NeurIPS 2024
>
> [ref2] Chain-of-Verification Reduces Hallucination in Large Language Models, ACL 2024
>
> [ref3] HalluMeasure: Fine-grained Hallucination Measurement Using Chain-of-Thought Reasoning, EMNLP 2024

---

> > ### Comment · Reviewer_CSJu · 2025-08-06
> >
> > Dear authors, I really appreciate your hard work on the response.
> >
> > ## Q1
> > I appreciate the hard work on the experiments for long video evaluation from LongVideoBench and VideoMME. From the official learderboard of these 2 datasets (https://longvideobench.github.io/, https://video-mme.github.io/home_page.html), the reported numbers from the proposed method is much lower than the previous open sourced and published works. This difference prove what I doubt about the long video capability of the proposed method.
> >
> > ## Q2
> > I partially agree with the response that the fairness discussion about the comparison. I understand there are indeed some difference on the model configurations, but what I expect is the proposed method using a better configuration (larger resolution, more frames, more data) to reach the SoTA instead of limiting the capability and performance of the previous works. Also, besides ViLAMP, there are tons of the baselines need to compare, including LinVT, TPO, VideoLLaMA 3, etc. I think the SoTA performance is critical to make the work better.
> >
> > ## Q3.1 & Q3.2 & Q3.3
> >
> > Thanks for the response. It solves my concerns. But what is [ref5]?
> >
> > ## Q4
> >
> > I acknowledge the built dataset and the dataset generation pipeline is the main contribution to the community.
> >
> > ---
> >
> > I think the response has solved partial of my concerns, but leave the performance bottleneck on long-form videos. Based on this, I adjust my rating to borderline accept.

---

> ### Author Response · Authors · 2025-08-05
>
> Dear Reviewer CSJu,
>
> Thanks once again for your valuable comments on our submission. As the discussion phase is approaching its end, we would like to kindly confirm whether we have sufficiently addressed all of your concerns (or at least part of them), and if there are any additional ones. We sincerely hope that we will be able to use the remaining time to engage in an open dialogue with domain experts to enhance the quality of our work.
>
> Thanks, Authors

---

> ### Author Response · Authors · 2025-08-07
>
> Dear Reviewer CSJu,
>
> _Sorry for the late reply, because we were preparing for some new results._
>
> We truly appreciate your valuable feedback, and your recognition of our contributions.  Particularly, thanks for your new inputs on the potential of our approach in long-form video reasoning, and we'd like to take this opportunity to examine it further.
>
> **First**, regarding the leaderboard, we note that the top-performing models generally fall into two categories:
> * **Large-scale models:** e.g., InternVL2.5-72B, InternVL2-40B, Qwen2-VL-72B, LLaVA-Video-72B, LLaVA-OneVision-72B, Aria (8x3.5B), etc.  These methods leverage significantly larger parameter sizes and benefit from much greater model capacity.
> * **Frame-/data-intensive 7B/8B models:** e.g., NVILA (1024 frames, over 100M samples), ViLAMP (up to 600 frames, 7.4M samples for pretraining and 1.8M data for fine-tuning), VideoChat-Flash (521 frames, 3.5M fine-tuning samples), VideoLLaMA 3 (180 frames, 37.54M samples for pretraining and 24.76M samples for fine-tuning), etc. These models use more frames and/or higher image resolutions, and often have access to larger training datasets. In contrast, our model is trained on only 16 frames and 102K samples for fine-tuning.
>
> These methods use stronger configuration (e.g., higher resolution, more frames, and more data) than us, and direct comparison is not completely fair.
>
> **Second**, while we agree with the reviewer that re-training with a stronger configuration would be valuable, it is currently beyond our computational budget and challenging within the short rebuttal window.
>
> Instead, here we consolidate the evaluate using **enhanced inference settings** with the **originally trained model**. Specifically, the enhanced inference settings are: input resolution is increased from  _256×28×28_ (~ 224×224) to _384×28×28_ (~ 320×320), and the number of frames is increased from _32_ to _128_. We compare our model against two suggested strong baselines, TPO and VideoLLaMA 3, on VideoMME:
>
> |Model|# Frames|Resolution|Accuracy (%)|
> |:-|:-:|:-:|:-:|
> |TPO|96|384×384|65.6|
> |VideoLLaMA 3|180|384×384|66.2|
> |VideoRFT|128|320×320|66.7|
>
> As seen, the results demonstrate that our method remains highly competitive, even trained in a more data-/compute-efficient manner.  It surpasses VideoLLaMA 3 (a generic video understanding model) by 0.5% and TPO (a model designed specifically for long-term videos) by 1.1% without re-training. In the next stage, we plan to fully retrain the model with the enhanced setup to further validate its potential.
>
> For the minor issue:  the citation [ref5] is a typo, and should be [ref3].
>
> Thanks again for your thoughtful feedback and constructive engagement.
>
> Best, Authors

---

### Official Review · Reviewer_xRp7 · 2025-07-05

**Clarity:** 3
**Significance:** 3
**Originality:** 3
**Rating:** 4
**Confidence:** 4

**Summary:**

The paper introduces VIDEORFT, a novel approach that extends the RFT paradigm to cultivate human-like video reasoning capabilities in MLLMs. The approach consists of a two-stage training paradigm: supervised fine-tuning with chain-of-thought annotations, followed by rule-based reinforcement learning, enhanced by a novel semantic-consistency reward. Recognizing the lack of high-quality video CoT datasets, the authors devise an automated cognitively inspired pipeline to generate rich, visually grounded CoT annotations and construct two large-scale datasets that support both SFT and RL phases. Extensive experiments across six benchmarks demonstrate the competitiveness and generalization ability of VideoRFT compared to both open-source and proprietary baselines.

**Questions:**

1. Regarding the details of the prompts, can the authors provide some specific examples of cognitive heuristic prompts? At the same time, I hope to discuss the universality of this method and its advantages over other methods.
2. Regarding the mitigation of visual hallucinations, how does the semantic consistency reward deal with situations where the language description does not completely match the visual information, or when the video evidence itself is ambiguous?

**Ethical Concerns:**

["NO or VERY MINOR ethics concerns only"]

**Limitations:**

yes

**Quality:**

3

**Strengths And Weaknesses:**

Strengths

1. Comprehensive Pipeline for CoT Data: This paper proposes a pipeline to solve the lack of high-quality video CoT data, while using structured video, cognitive-inspired CoT generation, and cross-modal optimization. The overall process ensures text-visual consistency on the basis of generating CoT.
2. Rich experimental results: This paper conducts a comprehensive comparison on six representative video reasoning benchmarks. VideoRFT performs well compared with most contemporary models, and the ablation results also clearly illustrate the impact of training and rule reward design.
3. Semantic consistency reward: A new reward mechanism is introduced to encourage the video description generated by the model to be consistent with visual evidence, helping to improve the shortcomings of the current video reasoning model.

Weaknesses
1. Innovative comparison with contemporaneous work: The unique innovations of VideoRFT in theory and method have not been fully discussed, and most of its specific contributions are more reflected in incremental combinations and improvements rather than in the proposal of new concepts.
2. Insufficient discussion of datasets: There is limited discussion on the diversity of automatically generated datasets, and the analysis is mainly focused on vocabulary and statistics, lacking in-depth analysis of the special data type of chain thinking (CoT).
3. Lack of details in dataset construction: The lack of specific examples of cognitive inspiration prompt strategies and cross-modal optimization processes in the text makes it difficult to understand and evaluate related methods.
4. Evaluation result analysis: Although this method has improved performance in most contemporaneous works, it has no significant advantage over Video-R1, suggesting that further consideration should be given to the validity of the data and its actual contribution to model performance.

---

> ### Author Rebuttal · Authors · 2025-07-31
>
> Thanks for the valuable feedback.
> To begin with, we focus on the following comment:
> >Q1.The unique innovations of VideoRFT in theory and method have not been fully discussed, and most of its specific contributions are more reflected in incremental combinations and improvements rather than in the proposal of new concepts.
>
> Thanks for comments. We clarify that our work contributes **a novel paradigm of reinforced fine-tuning** tailored for video reasoning, aiming at endowing MLLMs with internalized reasoning capabilities. Unlike prior works that rely on emergent abilities or heuristic prompt(e.g."Let think step by step").
>
> While we acknowledge that several contemporaneous works share a similar motivation, these methods were publicly released only after March 1st, 2025 (Video-R1 on March 27, and TinyLLaVA-Video-R1 on April 13). Per NeurIPS 2025 guidelines, *submissions will not be rejected on the basis of comparisons to contemporaneous work.* Nevertheless, we emphasize that our method contains several unique contributions:
>
> * We develop a novel CoT prompting scheme grounded in cognitive theory, which decomposes reasoning into five structured stages—**observation, task understanding, visual focus, reasoning, and reflection**. This mirrors human cognition and enables more interpretable and human-aligned CoTs generation than others. Additionally, we introduce a cross-modal refinement module to revise CoTs with visual evidence, reducing hallucination in CoTs by 28.5% and improving performance by +6.8% on average across datasets.
>
> * We propose a new reward function-**gated semantic-consistency reward**. It computes semantic alignment between video-grounded segments of CoTs and video input, and it uses a gating mechanism to activate the reward only when answer is right, avoiding reward of incorrect reasoning. To our knowledge, it's novel for video reasoning.
>
> We'll incorporate all discussions in next vision.
>
> >Q2.There is limited discussion on the diversity of automatically generated datasets, and the analysis is mainly focused on vocabulary and statistics, lacking in-depth analysis of the special data type of CoT.
>
> As requested, we provide a more in-depth analysis of our CoT data. Compared with standard QA annotations, CoTs are multi-step reasoning traces where each step must be both factual and logically coherent. Therefore, evaluating CoT quality requires examining two key properties: factual correctness(i.e., hallucination) and reasoning logicality.
>
> 1) **Hallucination Rate**: following [ref1], we randomly sample 200 CoTs to assess hallucination. We use GPT-4o to extract all referenced entities and actions from CoTs and verify whether they are grounded in the corresponding video/image. Ungrounded elements are manually reviewed. This result reveals our CoT annotations exhibit a low hallucination rate of 4.5%.
>
> 2) **Reasoning Logicality**: To further analyze the logicality of CoTs, we compare them with those generated by a contemporaneous method(Video-R1) on the same VQA instance:
>
> |Question|Video-R1|Ours|
> |-|-|-|
> |What did the person do to the laptop before lying on the bed? A.Opened B.Washed. C.Tidied up. D.Closed. *Ground Truth*: A|The person is lying on the bed in the beginning of the video. Then, they get up and start moving around the room. Hmm, what do they do next? They approach the computer desk and start interacting with the computer. **Wait, is there any bag, table, laptop, or phone/camera involved here? Oh, I see, they are not interacting with any of those objects directly.** They are just using the computer. Let‘s break it down further. The question asks ***what happened after they lied on the bed.*** They got up and used the computer. **There is no action of opening a bag, tidying up a table, putting down a laptop, or putting down a phone/camera.**|The question is asking what the person did to the laptop before lying on the bed. The video begins with **the person already lying on the bed, with the laptop screen glowing, indicating that the computer was already on**. The person then shifts and gets up to interact with the laptop. ***Given that the laptop is active from the start, it's clear that it was opened before the video began.*** Considering the options: A. Opened: **This aligns with the observation** that the laptop was already active when the person was lying on the bed. ...(*Analysis of other answers*)... The answer is most likely A.|
>
> As shown, the CoT from Video-R1 tends to drift toward unrelated details (e.g., bags, tables) and misunderstands the temporal cue in the question (ask what happened before lying on the bed, not after).
>
> In contrast, **our CoT follows a coherent, visually grounded reasoning**: It anchors the scene ("laptop screen glowing"), infers the temporal relationship, and eliminates wrong options through visual evidence. This reflects a logically clear and cognitively aligned reasoning process, closer to how humans analyze visual scenes to reason.
>
> >Q3.1.Lack of specific examples of cognitive inspiration prompt strategies and cross-modal optimization processes.
>
> >Q3.2.Can authors provide some specific examples of cognitive prompts? Discuss universality of method and advantages over other methods.
>
> 1) Sorry for the confusion. We clarify that ALL prompts used in the data generation process are presented in **[Supplementary Sec. B]**. To ease understanding, we copy key parts of prompts here.
>
> **The five sub-prompts for cognitive CoT generation in Eq. 1.**
> > Follow the multi-step reasoning approach:
> > 1. Simulate Watching: Imagine you are watching the entire video from beginning to end.
> > 2. Understand Question: Reflect on what the user is asking. Think carefully about what kind of answer is needed.
> > 3. Localize Relevant Moments: Consider which parts of the video are most related to the question. Focus on those segments in mental replay.
> > 4. Visual Reasoning: Describe what you "see" in those segments using natural visual language. Analyze the visual content to build answer.
> > 5. Answer Thoughtfully: Provide a clear and direct answer. Ensure reasoning is consistent with the video.
>
> **Part of prompt for cross-modal refinement**:
> > Your task is to revise hallucinations and errors in the CoT based on the visual content of the provided video.
> > 1. Carefully examine video, question, and CoT
> > 2. Identify only the reasoning steps that directly conflict with what is visually shown in the video:
> > 3. Correct CoT parts that contradict the video.
> > 4. Replace any text-based references with direct visual observations.
> > 5. Replace specific timestamps with broader temporal phrases.
>
> 2) **Universality and advantages**:
>
> **Regarding universality**: Our prompting is **task-agnostic**, making it broadly applicable across diverse video reasoning scenarios. During data generation, we employ the same prompting strategy across diverse QA taks (i.e., multiple-choice, open-ended QA, OCR) and task categories (i.e., general understanding, spatiotemporal analysis, scientific reasoning, mathematical problem, and document). This shows the strong universality.
>
> **Regarding advantages**: Our method integrates complementary strengths: the visual grounding of MLLMs and the deep reasoning capabilities of reasoning LLMs. Guided by cognitively inspired prompts, the LLM generates a human-like CoT, which is then refined by a MLLM with full video access. In contrast, prior methods either use LLMs with textual visual inputs, leading to hallucinations, or rely on MLLMs to mimic reasoning via surface-level prompts, lacking internalized reasoning modules.
>
> > Q4.Although this method has improved performance in most contemporaneous works, it has no significant advantage over Video-R1, suggesting that further consideration should be given to the validity of the data and its actual contribution to model performance.
>
> 1) **Performance**: **First**, we clarify that our approach shows better overall performance over Video-R1. Particularly, designed for video reasoning, more pronounced improvements on reasoning-intensive benchmarks. For instance, VideoRFT surpasses Video-R1 by +1.0% on VSI-Bench and +4.7% on MMVU. **Second**, to further address your concern, we extend the evaluation to include 5 additional video reasoning datasets. As seen, VideoRFT consistently outperforms Video-R1 by clear margins. This further confirms the superior performance of VideoRFT.
>
> |Model|LongVideo Bench|IntentQA|Next-QA|Causal-VidQA|DarkEventInfer|
> |:-|:-:|:-:|:-:|:-:|:-:|
> |Video-R1|50.2|88.2|78.3|61.7|56.0|
> |VideoRFT|**54.7**|**92.7**|**80.1**|**71.6**|**67.0**|
>
> 2) **Data Contribution**:
> To assess the actual impact of our dataset, We re-trained Qwen2.5-VL on both Video-R1 and our CoT dataset (with identical settings). Despite using only 62% samples (102K vs 165K), ours achieves better or comparable performance across:
>
> |Model|VSI.|VideoMMMU|MMVU|MV.|Temp.|VideoMME|
> |-|-|-|-|-|-|-|
> |Qwen2.5-VL + Video-R1 dataset|31.8|**47.4**|61.3|59.4|69.2|52.8|
> |Qwen2.5-VL + VideoRFT dataset|**32.7**|47.1|**63.5**|**59.9**|**69.6**|**53.0**|
>
> > Q5.How does the semantic consistency reward deal with situations where the language description doesn't completely match the visual information, or when the video evidence itself is ambiguous?
>
> To clarify, we restate reward formulation below:
>
> $R_s=\min(1,w\times \max(\text{cos}(t,v),0)) \quad R=R_f+R_a+\mathbb{1}[R_a>0]\cdot R_s$
>
> 1) Regarding "language description $t$ doesn't completely match the visual information $s$". If completely unrelated, $\cos(t,v)<0$ yields $R_s=0$, contributing no reward signal. If weakly related, $\cos(t, v)$ is a small positive value, and $R_s=w \times \cos(t,v)$. As training proceeds, alignment improves and $R_s$ gradually increases.
>
> 2) When the **video evidence is ambiguous**, the model might tend to give a wrong answer, i.e., $R_a=0$. Thus, $R_s$ is excluded from the final reward.
>
> [ref1] HalluMeasure: Fine-grained Hallucination Measurement Using Chain-of-Thought Reasoning, EMNLP2024

---

> > ### Comment · Reviewer_xRp7 · 2025-08-05
> > **noted**
> >
> > I have reviewed the results you provided, and I would like to confirm that they have addressed my concerns.

---

> ### Author Response · Authors · 2025-08-05
>
> Dear Reviewer xRp7,
>
> Thanks once again for your valuable comments on our submission. As the discussion phase is approaching its end, we would like to kindly confirm whether we have sufficiently addressed all of your concerns (or at least part of them), and if there are any additional ones. We sincerely hope that we will be able to use the remaining time to engage in an open dialogue with domain experts to enhance the quality of our work.
>
> Thanks, Authors

---

### Note · Authors · 2025-08-13

Dear AC and Reviewers:

Thanks again for your valuable time and dedication to our submission. We appreciate the opportunity to provide the final remarks.

VideoRFT is an **early work exploring the reinforced fine-tuning (RFT)** paradigm for video reasoning, featuring a **multi-expert, cognition-inspired pipeline** that addresses the scarcity of video CoT data and a **novel reward function** tailored to video reasoning. Reviewers recognized **our pipeline as a notable innovation** (3mwX), **our dataset as a valuable contribution** (CSJu, 3mwX), **our reward as novel** (xRp7, 3mwX, 9Vu5), and **our performance as strong** (xRp7, 9Vu5).

During the rebuttal stage, we addressed the main concerns raised by the reviewers, including but not limited to **performance comparisons with contemporaneous work**, **CoT quality**, **long-form video reasoning capability**, and **the efficiency of our reward function**. We are encouraged that these clarifications have led to potential score increases from reviewers xRp7, CSJu, and 9Vu5.

We would like to take this opportunity to reiterate two points, although they have already been addressed in the rebuttal. First, regarding Reviewer **CSJu’s concern on the performance bottleneck for long-form videos**: We consolidated additional evaluations using enhanced inference settings (~320×320 resolution and 128 frames) with the originally trained model and found that our method surpasses VideoLLaMA 3 and TPO (a model designed for long videos), demonstrating **strong long-form video reasoning capability** in both a **data- and compute-efficient** manner. Second, regarding Reviewer **3mwX’s concern on the runtime overhead of the SigLIP module**: we performed an ablation on reward computation and reported the actual overhead in the rebuttal. During the discussion stage, we corrected a **minor unit conversion oversight** in those results. The updated measurement shows only a **4.4% runtime overhead** (1.15s/26.30s). We believe the issues have been properly addressed.

We will follow reviewers' comments to revise the article, incorporating all new results and discussions.

Best, Authors

---

### Decision · Program_Chairs · 2025-09-17

**Decision:**

Accept (poster)

**Comment:**

The manuscript investigates reinforcement fine-tuning (RFT) for video reasoning. The proposed approach, named VideoRFT, follows the two-stage scheme in RFT (SFT with CoT and online RL). To this end, the manuscript builds a an automatic CoT curation pipeline, resulting in two new datasets: VideoRFT-CoT-102K for SFT and VideoRFT-RL-310K for RL. To consolidate the RL phase, the manuscript introduces a semantic-consistency reward that aims to improve the alignment between textual reasoning and visual evidence. The manuscript received ratings of three borderline accept and one borderline reject. Reviewers appreciated the proposed dataset and automated pipeline for generating CoT data, semantic-consistency reward. Reviewers also raised several questions including, the need for fine-tuning compared to prompting, the generalization ability of the fine-tuned model, and computational overhead during training due to SigCLIP embeddings. Post-rebuttal, most reviewers expressed thaat the initial concerns are generally resolved (e.g., questions regarding fine-tuning compared to prompting, generalization capabilities of fine-tuned model), but Reviewer 3mwX still questioned the provided computational overhead comparison with SigCLIP embeddings. This was addressed in the follow-up response from the authors, where they provided rectified computational overhead (e.g., logs) and showed no significant computational overhead. Given that three reviewers are positive, the provided rebuttal, and author-reviewer discussions, the recommendation is accept.